# Learning Energy-based Model via Dual-MCMC Teaching

**Jiali Cui, Tian Han**
Department of Computer Science, Stevens Institute of Technology
{jcui7,than6}@stevens.edu

## Abstract

This paper studies the fundamental learning problem of the energy-based model (EBM). Learning the EBM can be achieved using the maximum likelihood estimation (MLE), which typically involves the Markov Chain Monte Carlo (MCMC) sampling, such as the Langevin dynamics. However, the noise-initialized Langevin dynamics can be challenging in practice and hard to mix. This motivates the exploration of joint training with the generator model where the generator model serves as a complementary model to bypass MCMC sampling. However, such a method can be less accurate than the MCMC and result in biased EBM learning. While the generator can also serve as an initializer model for better MCMC sampling, its learning can be biased since it only matches the EBM and has no access to empirical training examples. Such biased generator learning may limit the potential of learning the EBM. To address this issue, we present a joint learning framework that interweaves the maximum likelihood learning algorithm for both the EBM and the complementary generator model. In particular, the generator model is learned by MLE to match both the EBM and the empirical data distribution, making it a more informative initializer for MCMC sampling of EBM. Learning generator with observed examples typically requires inference of the generator posterior. To ensure accurate and efficient inference, we adopt the MCMC posterior sampling and introduce a complementary inference model to initialize such latent MCMC sampling. We show that three separate models can be seamlessly integrated into our joint framework through two (dual-) MCMC teaching, enabling effective and efficient EBM learning.

## 1 Introduction

Deep generative models have made significant progress in learning complex data distributions [35, 18, 37, 34, 19, 22, 5] and have found successful applications in a wide range of real-world scenarios [26, 6, 11, 15]. Among these, the energy-based model (EBM) [5, 6, 29, 8, 39, 2, 3] has gained particular interest as a flexible and expressive generative model with an energy function parameterized by a neural network. Learning the EBM can be accomplished via the maximum likelihood estimation (MLE), which involves the Markov Chain Monte Carlo (MCMC) sampling in high-dimensional data space. However, such MCMC sampling has shown to be challenging [30, 7, 33, 13], as it may take a long time to mix between different local modes with a non-informative noise initialization [43, 15].

To address this challenge, recent advances have explored employing complementary models to substitute for MCMC sampling [15, 16, 12, 17, 24]. One notable example is the generator model. The generator model incorporates a top-down generation network that is capable of mapping low-dimensional latent space to high-dimensional data space and admits efficient sample generation. The generator is learned to match the EBM so that MCMC sampling can be replaced by generator ancestral

37th Conference on Neural Information Processing Systems (NeurIPS 2023).

sampling. However, such direct generator sampling has shown to be less accurate and suboptimal [43]. To alleviate this issue, [42, 43] introduced cooperative learning, where samples generated by the generator model serve as initial points, and then followed by a finite-step MCMC revision process. While this gradient-based MCMC revision process can be more accurate, the generator model learned relies solely on the EBM and has no access to the observed empirical observations. As a result, this learning scheme may render biased generator learning, which in turn caps the potential of learning a strong EBM. An effective joint learning scheme for the EBM and its complementary generator model is needed, yet still in its infancy.

In this paper, we present a novel learning scheme that can seamlessly integrate the EBM and complementary models into a joint probabilistic framework. Specifically, both the EBM and complementary generator model are learned to match the empirical data distribution, while the generator model, at the same time, is also learned to match the EBM. Learning the generator model with empirical training examples can be achieved with MLE, which typically requires access to the generator posterior as an inference process. To ensure an effective and efficient inference, we employ the MCMC posterior sampling with the complementary inference model learned as an initializer. Together with MCMC sampling of EBM being initialized by the generator model, such two MCMC samplings can be further used as two MCMC revision processes that *teach* the generator and inference model to absorb MCMC-revised samples, thus we term our framework *dual-MCMC teaching*. We show that our joint framework is capable of *teaching* the complementary models and thus learning a strong EBM.

Our contributions can be summarized as follows:

- We introduce a novel method that integrates the EBM and its complementary models into a joint learning scheme.
- We propose the use of *dual-MCMC teaching* for generator and inference models to facilitate efficient yet accurate sampling and inference, which in turn leads to effective EBM learning.
- We conduct extensive experiments to demonstrate the superior performance of our EBM.

## 2 Preliminary

### 2.1 Energy-based Model

Let $\mathbf{x} \in R^D$ be the high-dimensional observed examples. The energy-based model (EBM) [41, 6, 5, 29] represents data uncertainty with an undirected probability density defined as

$$\pi_\alpha(\mathbf{x}) = \frac{1}{Z(\alpha)} \exp\left[f_\alpha(\mathbf{x})\right], \tag{1}$$

where $-f_\alpha(\mathbf{x})$ is the energy function parameterized with parameters $\alpha$, and $Z(\alpha)$ ($= \int_\mathbf{x} \exp[f_\alpha(\mathbf{x})]d\mathbf{x}$) is the partition function or normalizing constant.

**Maximum likelihood estimation.** The maximum likelihood estimation (MLE) is known for being an asymptotically optimal estimator and can be used for training the EBM. In particular, with observed examples, $\{\mathbf{x}^{(i)}, i = 1, 2, ..., n\}$, the MLE learning of EBM maximizes the log-likelihood $L_\pi(\alpha) = \frac{1}{n}\sum_{i=1}^n \log \pi_\alpha(\mathbf{x}^{(i)})$. If the sample size $n$ is large enough, the maximum likelihood estimator minimizes the $\mathrm{KL}(p_\mathrm{d}(\mathbf{x})\|\pi_\alpha(\mathbf{x}))$ which is the Kullback-Leibler (KL) divergence between the empirical data distribution $p_\mathrm{d}(\mathbf{x})$ and the EBM distribution $\pi_\alpha(\mathbf{x})$. The gradient $\frac{\partial}{\partial\alpha}L_\pi(\alpha)$ is computed as

$$\max_\alpha L_\pi(\alpha) = \min_\alpha \mathrm{KL}(p_\mathrm{d}(\mathbf{x})\|\pi_\alpha(\mathbf{x})), \quad \text{where}$$

$$\frac{\partial}{\partial\alpha}L_\pi(\alpha) = \mathbb{E}_{p_\mathrm{d}(\mathbf{x})}[\frac{\partial}{\partial\alpha}f_\alpha(\mathbf{x})] - \mathbb{E}_{\pi_\alpha(\mathbf{x})}[\frac{\partial}{\partial\alpha}f_\alpha(\mathbf{x})] \tag{2}$$

Given such a gradient, the EBM can be learned via stochastic gradient ascent.

**Sampling from EBM.** The Eqn.2 requires sampling from the EBM $\pi_\alpha(\mathbf{x})$, which can be achieved via Markov Chain Monte Carlo (MCMC) sampling, such as the Langevin dynamics [28]. Specifically, to sample from the EBM, the Langevin dynamics iteratively updates as

$$\mathbf{x}_{\tau+1} = \mathbf{x}_\tau + s\frac{\partial}{\partial\mathbf{x}_\tau}\log \pi_\alpha(\mathbf{x}_\tau) + \sqrt{2s}U_\tau \tag{3}$$

where $\tau$ indexes the time step, $s$ is the step size and $U_\tau \sim \mathcal{N}(0, I_D)$.

As $s \to 0$, and $\tau \to \infty$, the distribution of $\mathbf{x}_\tau$ will converge to the target distribution $\pi_\alpha(\mathbf{x}_\tau)$ regardless of the initial distribution of $\mathbf{x}_0$ [28]. The existing practice [29, 30, 5] adopts non-informative distribution for $\mathbf{x}_0$, such as unit Gaussian or uniform, to initialize the Langevin transition, but it can be extremely inefficient and ineffective as they usually take a long time to converge between different modes and are also non-stable in practice [43]. The ability to generate efficient and effective samples from the model distribution becomes the key step toward training successful EBMs. In this paper, we study the complementary model, i.e., *generator model*, as an informative initializer for effective yet efficient MCMC exploration toward better EBM training.

## 2.2  Generator Model

Various works [15, 12, 17, 24] have explored the use of the generator as an amortized sampler to replace the costly noise-initialized MCMC sampling for EBM training. Such a learning approach relies on samples directly drawn from complementary models, which can be less accurate than iterative MCMC sampling as it lacks a fine-grained exploration of the energy landscape. [42, 43] propose the cooperative scheme in which MCMC sampling of EBM is initialized by the generated samples from the generator model. However, the generator model has no access to the observed training examples, and such a *biased* generator learning makes the EBM sampling ineffective and renders limited model training.

In this paper, the generator model is learned by MLE to match both the empirical data distribution and the EBM distribution. Such training ensures a stronger generator model which will further facilitate a more effective EBM sampling and learning. We present the background of the generator model and its MLE learning algorithm below, which shall serve as the foundation of our proposed method.

**Generator model.** Let $\mathbf{z} \in R^d$ $(d < D)$ be the low-dimensional latent variables. The generator model [14, 10, 21] seeks to explain the observation signal $\mathbf{x}$ by a latent vector $\mathbf{z}$ and can be specified as

$$p_\theta(\mathbf{x}, \mathbf{z}) = p(\mathbf{z})p_\theta(\mathbf{x}|\mathbf{z}) \tag{4}$$

where $p(\mathbf{z})$ is a known prior distribution such as unit Gaussian, e.g., $p(\mathbf{z}) \sim \mathcal{N}(0, I_d)$, and $p_\theta(\mathbf{x}|\mathbf{z}) \sim \mathcal{N}(g_\theta(\mathbf{z}), \sigma^2 I_D)$ is the generation model that is specified by neural network $g_\theta(.)$ that maps from latent space to data space.

**Maximum likelihood estimation.** The MLE learning of the generator model computes log-likelihood over the observed examples as $L_p(\theta) = \frac{1}{n}\sum_{i=1}^n \log p_\theta(\mathbf{x}^{(i)})$, where $p_\theta(\mathbf{x})(= \int_\mathbf{z} p_\theta(\mathbf{x}, \mathbf{z})d\mathbf{z})$ is the marginal distribution. If the sample size $n$ is large, it is equivalent to minimizing the KL divergence $\mathrm{KL}(p_\mathrm{d}(\mathbf{x})\|p_\theta(\mathbf{x}))$. The gradient of the likelihood $L_p(\theta)$ can be obtained via:

$$\max_\theta L_p(\theta) = \min_\theta \mathrm{KL}(p_\mathrm{d}(\mathbf{x})\|p_\theta(\mathbf{x})), \quad \text{where}$$

$$\frac{\partial}{\partial\theta}L_p(\theta) = \mathbb{E}_{p_\mathrm{d}(\mathbf{x})p_\theta(\mathbf{z}|\mathbf{x})}\big[\frac{\partial}{\partial\theta}\log p_\theta(\mathbf{x}, \mathbf{z})\big] \tag{5}$$

With such a gradient, the generator model can be learned via stochastic gradient ascent.

**Sampling from generator posterior.** The Eqn.5 requires the sampling from the generator posterior $p_\theta(\mathbf{z}|\mathbf{x})$. One can use MCMC sampling such as Langevin dynamics [28] that iterates

$$\mathbf{z}_{\tau+1} = \mathbf{z}_\tau + s\frac{\partial}{\partial\mathbf{z}_\tau}\log p_\theta(\mathbf{z}_\tau|\mathbf{x}) + \sqrt{2s}U_\tau \tag{6}$$

where $U_\tau \sim \mathcal{N}(0, I_d)$. Such a Langevin process is an explaining-away inference where the latent factors compete with each other to explain each training example. As $s \to 0$, and $\tau \to \infty$, the distribution of $\mathbf{z}_\tau$ will converge to the posterior $p_\theta(\mathbf{z}|\mathbf{x})$ regardless of the initial distribution of $\mathbf{z}_0$ [28]. However, noise-initialized Langevin [14, 31] can be ineffective in traversing the latent space and hard to mix. In this paper, we introduce a complementary model, i.e., *inference model*, as an informative initializer for effective yet efficient latent space MCMC exploration for better generator and EBM training.

## 2.3 Inference model

The inference model $q_\phi(\mathbf{z}|\mathbf{x})$ is adopted in VAEs [21, 32] as an amortized sampler to bypass the costly noise-initialized latent space MCMC sampling. In VAEs, $q_\phi(\mathbf{z}|\mathbf{x})$ is Gaussian parameterized, i.e., $\mathcal{N}(\mu_\phi(\mathbf{x}), V_\phi(\mathbf{x}))$, where $\mu_\phi(\mathbf{x})$ is the mean $d$-dimensional mean vector and $V_\phi(\mathbf{x})$ is the $d$-dimensional diagonal covariance matrix. Such a Gaussian parameterized inference model is a tractable approximation to the true generator posterior $p_\theta(\mathbf{z}|\mathbf{x})$, but can be limited to approximate the multi-modal posterior. We adopt the same Gaussian parametrization of $q_\phi(z|x)$ in this paper, but unlike the VAEs, our inference model serves as an initializer network that jump-starts the latent MCMC sampling from an informative initialization. The marginal distribution obtained after Langevin can be more general and multi-modal than the Gaussian distribution.

## 3 Methodology

To effectively learn the EBM, we propose a joint learning framework that interweaves maximum likelihood learning algorithms for both the EBM and its complementary models. For the MLE learning of the EBM, MCMC sampling can be initialized through the complementary generator model, while for the MLE learning of the generator model, the latent MCMC sampling can be initialized by the complementary inference model. Three models are seamlessly integrated into our joint framework and are learned through *dual-MCMC teaching*.

### 3.1 Dual-MCMC Sampling

The EBM $\pi_\alpha(\mathbf{x})$, generator $p_\theta(\mathbf{x})$ and the inference model $q_\phi(\mathbf{z}|\mathbf{x})$ defined in Sec.2 naturally specify the three densities on joint space $(\mathbf{x}, \mathbf{z})$, i.e.,

$$P_\theta(\mathbf{x}, \mathbf{z}) = p_\theta(\mathbf{x}|\mathbf{z})p(\mathbf{z}), \quad \Pi_{\alpha,\phi}(\mathbf{x}, \mathbf{z}) = \pi_\alpha(\mathbf{x})q_\phi(\mathbf{z}|\mathbf{x}), \quad Q_\phi(\mathbf{x}, \mathbf{z}) = p_\mathrm{d}(\mathbf{x})q_\phi(\mathbf{z}|\mathbf{x})$$

The *generator density* $P_\theta(\mathbf{x}, \mathbf{z})$ specifies the joint density through ancestral generator sampling from prior latent vectors. Both the *joint EBM density* $\Pi_{\alpha,\phi}(\mathbf{x}, \mathbf{z})$ and *data density* $Q_\phi(\mathbf{x}, \mathbf{z})$ include inference model $q_\phi(\mathbf{z}|\mathbf{x})$ to bridge the marginal distribution to joint $(\mathbf{x}, \mathbf{z})$ space. However, $q_\phi(\mathbf{z}|\mathbf{x})$ is modeled and learned from two different perspectives, one on empirical observed data distribution $p_\mathrm{d}(\mathbf{x})$ for *real data inference*, and one on EBM density $\pi_\alpha(\mathbf{x})$ for *generated sample inference*.

The joint learning schemes [15, 12, 17, 24] based on these joint distributions can be limited, because 1) the generator samples from $P_\theta(\mathbf{x}, \mathbf{z})$ is conditionally Gaussian distributed (Sec.2.2) which can be ineffective to capture the high-dimensional multi-modal empirical data distribution, and 2) the inference model $q_\phi(\mathbf{z}|\mathbf{x})$ on observed training examples is assumed to be conditionally Gaussian distributed (Sec.2.3) that is incapable of explaining-away inference [14].

To address the above limitations, we introduce two joint distributions that incorporate MCMC sampling as revision processes,

$$\tilde{P}_{\theta,\alpha}(\mathbf{x}, \mathbf{z}) = \mathcal{T}_\alpha^\mathbf{x} p_\theta(\mathbf{x}|\mathbf{z})p(\mathbf{z}) \quad \tilde{Q}_{\phi,\theta}(\mathbf{x}, \mathbf{z}) = p_\mathrm{d}(\mathbf{x})\mathcal{T}_\theta^\mathbf{z} q_\phi(\mathbf{z}|\mathbf{x})$$

where $\mathcal{T}_\theta^\mathbf{z}(\cdot)$ denotes the Markov transition kernel of finite step Langevin dynamics that samples $\mathbf{z}$ from $p_\theta(\mathbf{z}|\mathbf{x})$ (see Eqn.6), and $\mathcal{T}_\alpha^\mathbf{x}(\cdot)$ denotes the transition kernel that samples $\mathbf{x}$ from $\pi_\alpha(\mathbf{x})$ as shown in Eqn.3. Therefore, $\mathcal{T}_\alpha^\mathbf{x} p_\theta(\mathbf{x})(= \int_{\mathbf{x}'} \int_\mathbf{z} \mathcal{T}_\alpha^\mathbf{x}(\mathbf{x}')p_\theta(\mathbf{x}', \mathbf{z})d\mathbf{z}d\mathbf{x}')$ indicates the marginal distribution of $\mathbf{x}$ obtained by running MCMC transition $\mathcal{T}_\alpha^\mathbf{x}(\cdot)$ that is initialized from $p_\theta(\mathbf{x})$. Similarly, $\mathcal{T}_\theta^\mathbf{z} q_\phi(\mathbf{z}|\mathbf{x})$ represents the marginal distribution of $\mathbf{z}$ obtained by running $\mathcal{T}_\theta^\mathbf{z}(\cdot)$ that is initialized from $q_\phi(\mathbf{z}|\mathbf{x})$ given observation $\mathbf{x}$ (i.e., $\mathcal{T}_\theta^\mathbf{z} q_\phi(\mathbf{z}|\mathbf{x}) = \int_{\mathbf{z}'} \mathcal{T}_\theta^\mathbf{z}(\mathbf{z}')q_\phi(\mathbf{z}'|\mathbf{x})d\mathbf{z}')$.

The $\tilde{P}_{\theta,\alpha}(\mathbf{x}, \mathbf{z})$, as a *revised generator density*, is more expressive on $\mathbf{x}$-space than $P_\theta(\mathbf{x}, \mathbf{z})$ as the generated samples from $p_\theta(\mathbf{x})$ are refined via the EBM-guided MCMC sampling. $\tilde{Q}_{\phi,\theta}(\mathbf{x}, \mathbf{z})$, as a *revised data density*, can be more expressive on $\mathbf{z}$-space than $Q_\phi(\mathbf{x}, \mathbf{z})$, as the latent samples from $q_\phi(\mathbf{z}|\mathbf{x})$ are revised via the generator-guided explaining-away MCMC inference. These MCMC-revised joint densities will be used for better EBM training, while at the same time, they will guide and *teach* the generator and inference model to better initialize and facilitate MCMC samplings.

We jointly train three models within a probabilistic framework based on KL divergence between joint densities. We present below our learning algorithm in an alternative and iterative manner where the new model parameters are updated based on the current model parameters. We present the learning algorithm in Appendix.

## 3.2 Learning Energy-based Model

Learning the EBM is based on the minimization of KL divergences as

$$\min_{\alpha} D_\pi(\alpha) = \min_{\alpha} \mathrm{KL}(\tilde{Q}_{\phi_t,\theta_t}(\mathbf{x},\mathbf{z})\|\Pi_{\alpha,\phi}(\mathbf{x},\mathbf{z})) - \mathrm{KL}(\tilde{P}_{\theta_t,\alpha_t}(\mathbf{x},\mathbf{z})\|\Pi_{\alpha,\phi}(\mathbf{x},\mathbf{z}))$$

$$\text{where} \quad -\frac{\partial}{\partial\alpha}D_\pi(\alpha) = \mathbb{E}_{p_d(\mathbf{x})}[\frac{\partial}{\partial\alpha}f_\alpha(\mathbf{x})] - \mathbb{E}_{\mathcal{T}_{\alpha_t}^{\mathbf{x}}p_{\theta_t}(\mathbf{x})}[\frac{\partial}{\partial\alpha}f_\alpha(\mathbf{x})] \tag{7}$$

where $\alpha_t, \theta_t, \phi_t$ denote fixed copies of EBM, generator, and inference model at the $t$-th step in an iterative algorithm. The joint densities $\tilde{Q}_{\phi_t,\theta_t}$ and $\tilde{P}_{\theta_t,\alpha_t}$ are based on this current iteration.

Comparing Eqn.7 to Eqn.2, we compute sampling from EBM through Langevin transition with current $p_{\theta_t}(\mathbf{x})$ as an initializer, i.e., $\mathcal{T}_{\alpha_t}^{\mathbf{x}}p_{\theta_t}(\mathbf{x})$. Such a generator initialized MCMC is more effective and efficient compared to the noise-initialized transition, $\mathcal{T}_{\alpha_t}^{\mathbf{x}}(\epsilon_{\mathbf{x}})$ where $\epsilon_{\mathbf{x}} \sim \mathcal{N}(0, I_D)$, that is used in recent literature [29, 5, 6].

**MLE perturbation.** The above joint space KL divergences are equivalent to the marginal version,

$$\min_{\alpha} D_\pi(\alpha) = \min_{\alpha} \mathrm{KL}(p_d(\mathbf{x})\|\pi_\alpha(\mathbf{x})) - \mathrm{KL}(\mathcal{T}_{\alpha_t}^{\mathbf{x}}p_{\theta_t}(\mathbf{x})\|\pi_\alpha(\mathbf{x})) \tag{8}$$

Additionally, $\mathcal{T}_{\alpha_t}^{\mathbf{x}}p_{\theta_t}(\mathbf{x}) \to \pi_{\alpha_t}(\mathbf{x})$ if $s \to 0$ and $\tau \to \infty$ (see Eqn.3), thus Eqn.8 amounts to the approximation of MLE objective function with a KL perturbation term, i.e.,

$$\mathrm{KL}(p_d(\mathbf{x})\|\pi_\alpha(\mathbf{x})) - \mathrm{KL}(\pi_{\alpha_t}(\mathbf{x})\|\pi_\alpha(\mathbf{x})) \tag{9}$$

Such surrogate form is more tractable than the MLE objective function, since the $\log Z(\alpha)$ term is canceled out. The $\pi_\alpha(\mathbf{x})$ seeks to approach the data distribution $p_d(\mathbf{x})$ while escapes from its current version $\pi_{\alpha_t}(\mathbf{x})$, thus can be treated as its own critic. The learning of EBM can then be interpreted as a *self-adversarial learning* [15, 40].

**Connection to variational learning.** It is also tempting to learn the EBM without MCMC sampling via gradient $\mathbb{E}_{p_d(\mathbf{x})}[\frac{\partial}{\partial\alpha}f_\alpha(\mathbf{x})] - \mathbb{E}_{p_{\theta_t}(\mathbf{x})}[\frac{\partial}{\partial\alpha}f_\alpha(\mathbf{x})]$ (i.e., $\min_{\alpha} \mathrm{KL}(Q_{\phi_t}(\mathbf{x},\mathbf{z})\|\Pi_{\alpha,\phi}(\mathbf{x},\mathbf{z})) - \mathrm{KL}(P_{\theta_t}(\mathbf{x},\mathbf{z})\|\Pi_{\alpha,\phi}(\mathbf{x},\mathbf{z}))$ ), which underlies the variational joint learning [15, 4, 12, 24]. Compared to Eqn.7, their generator serves as a direct sampler for EBM, while we perform the EBM self-guided MCMC revision for more accurate samples.

## 3.3 Learning Generator Model via Dual-MCMC Teaching

As a complementary model for learning the EBM, the generator model becomes a key ingredient toward success. The generator model is learned through the minimization of KL divergences as

$$\min_{\theta} D_p(\theta) = \min_{\theta} \mathrm{KL}(\tilde{Q}_{\phi_t,\theta_t}(\mathbf{x},\mathbf{z})\|P_\theta(\mathbf{x},\mathbf{z})) + \mathrm{KL}(\tilde{P}_{\theta_t,\alpha_t}(\mathbf{x},\mathbf{z})\|P_\theta(\mathbf{x},\mathbf{z}))$$

$$\text{where} \quad -\frac{\partial}{\partial\theta}D_p(\theta) = \mathbb{E}_{p_d(\mathbf{x})\mathcal{T}_{\theta_t}^{\mathbf{z}}q_{\phi_t}(\mathbf{z}|\mathbf{x})}[\frac{\partial}{\partial\theta}\log p_\theta(\mathbf{x},\mathbf{z})] + \mathbb{E}_{\mathcal{T}_{\alpha_t}^{\mathbf{x}}p_{\theta_t}(\mathbf{x}|\mathbf{z})p(\mathbf{z})}[\frac{\partial}{\partial\theta}\log p_\theta(\mathbf{x},\mathbf{z})] \tag{10}$$

where both the $\tilde{Q}_{\phi_t,\theta_t}$ and $\tilde{P}_{\theta_t,\alpha_t}$ are based on the current iteration. The revised data density $\tilde{Q}_{\phi_t,\theta_t}$ *teaches* the generator to better match with empirical data observations through the first KL term, and the revised generator density $\tilde{P}_{\theta_t,\alpha_t}$ *teaches* the generator to better match with generated samples through the second KL term. As we describe below, such a joint minimization scheme provides a tractable approximation of the generator learning with marginal distribution, i.e.,

$$\min_{\theta} \mathrm{KL}(p_d(\mathbf{x})\|p_\theta(\mathbf{x})) + \mathrm{KL}(\pi_{\alpha_t}(\mathbf{x})\|p_\theta(\mathbf{x}))$$

where generator model $p_\theta(\mathbf{x})$ learns to match the $p_d(\mathbf{x})$ on empirical data observations and catch up with the current EBM density $\pi_{\alpha_t}(\mathbf{x})$ through guidance of its generated samples.

**MLE perturbation on $p_d$.** Our generator model matches empirical data distribution $p_d(\mathbf{x})$ through $\mathrm{KL}(\tilde{Q}_{\phi_t,\theta_t}(\mathbf{x},\mathbf{z})\|P_\theta(\mathbf{x},\mathbf{z}))$ and is equivalent to the marginal version that follows,

$$\mathrm{KL}(\tilde{Q}_{\phi_t,\theta_t}(\mathbf{x},\mathbf{z})\|P_\theta(\mathbf{x},\mathbf{z})) = \mathrm{KL}(p_d(\mathbf{x})\|p_\theta(\mathbf{x})) + \mathbb{E}_{p_d(\mathbf{x})}[\mathrm{KL}(\mathcal{T}_{\theta_t}^{\mathbf{z}}q_{\phi_t}(\mathbf{z}|\mathbf{x})\|p_\theta(\mathbf{z}|\mathbf{x}))] \tag{11}$$

Given $s \to 0$ and $\tau \to \infty$, $\mathcal{T}_{\theta_t}^{\mathbf{z}}q_{\phi_t}(\mathbf{z}|\mathbf{x}) \to p_{\theta_t}(\mathbf{z}|\mathbf{x})$ (see Eqn.6), the first KL term (in Eqn.10) thus approximates the true MLE objective function with additional KL perturbation term, i.e.,

$$\mathrm{KL}(p_d(\mathbf{x})\|p_\theta(\mathbf{x})) + \mathbb{E}_{p_d(\mathbf{x})}[\mathrm{KL}(p_{\theta_t}(\mathbf{z}|\mathbf{x})\|p_\theta(\mathbf{z}|\mathbf{x}))] \tag{12}$$

Such surrogate form in joint density upper-bounds (i.e., majorizes) the true MLE objective $\mathrm{KL}(p_{\mathrm{d}}(\mathbf{x})\|p_\theta(\mathbf{x}))$ and can be more tractable as it involves the complete-data model with latent vector $\mathbf{z}$ has been inferred in the current learning step. Minimizing the surrogate form in the iterative algorithm makes the generator $p_\theta(\mathbf{x})$ to be closer to the empirical $p_{\mathrm{d}}(\mathbf{x})$ due to its majorization property [15].

**MLE perturbation on $\pi_{\alpha_t}$.** Our generator model is learned to catch up with the EBM model $\pi_\alpha(\mathbf{x})$ through the second term $\mathrm{KL}(\tilde{P}_{\theta_t,\alpha_t}(\mathbf{x},\mathbf{z})\|P_\theta(\mathbf{x},\mathbf{z}))$. It is equivalent to the marginal version as

$$\mathrm{KL}(\tilde{P}_{\theta_t,\alpha_t}(\mathbf{x},\mathbf{z})\|P_\theta(\mathbf{x},\mathbf{z})) = \mathrm{KL}(\mathcal{T}^{\mathbf{x}}_{\alpha_t}p_{\theta_t}(\mathbf{x})\|p_\theta(\mathbf{x})) + \mathbb{E}_{\mathcal{T}^{\mathbf{x}}_{\alpha_t}p_{\theta_t}(\mathbf{x})}[\mathrm{KL}(p_{\theta_t}(\mathbf{z}|\mathbf{x})\|p_\theta(\mathbf{z}|\mathbf{x}))] \quad (13)$$

With $s \to 0$ and $\tau \to \infty$, $\mathcal{T}^{\mathbf{x}}_{\alpha_t}p_{\theta_t}(\mathbf{x}) \to \pi_{\alpha_t}(\mathbf{x})$ (see Eqn.3), our second KL term approximates (in Eqn.10) the MLE objective on $\pi_{\alpha_t}$ for generator,

$$\mathrm{KL}(\pi_{\alpha_t}(\mathbf{x})\|p_\theta(\mathbf{x})) + \mathbb{E}_{\pi_{\alpha_t}(\mathbf{x})}[\mathrm{KL}(p_{\theta_t}(\mathbf{z}|\mathbf{x})\|p_\theta(\mathbf{z}|\mathbf{x}))] \quad (14)$$

Such surrogate in joint density again upper-bounds (i.e., majorizes) the true MLE objective on generated samples, i.e., $\mathrm{KL}(\pi_{\alpha_t}(\mathbf{x})\|p_\theta(\mathbf{x}))$, and thus the generator $p_\theta(\mathbf{x})$ updates to be closer to the EBM $\pi_\alpha(\mathbf{x})$ at the current iteration.

**Connection to variational learning.** Without MCMC inference, the generator model can be learned with inference model to match the empirical data distribution, i.e., $\min_\theta \mathrm{KL}(Q_{\phi_t}(\mathbf{x},\mathbf{z})\|P_\theta(\mathbf{x},\mathbf{z}))$, which underlies VAEs [21, 32, 26, 37]. Compared to Eqn.11, VAEs seek to minimize $\mathrm{KL}(p_{\mathrm{d}}(\mathbf{x})\|p_\theta(\mathbf{x})) + \mathbb{E}_{p_{\mathrm{d}}(\mathbf{x})}[\mathrm{KL}(q_{\phi_t}(\mathbf{z}|\mathbf{x})\|p_\theta(\mathbf{z}|\mathbf{x}))]$ where $q_\phi(\mathbf{z}|\mathbf{x})$ is assumed to be Gaussian distributed which has limited capacity for generator learning.

**Connection to cooperative learning.** Cooperative learning schemes [42, 43] share similar EBM training procedures but can be fundamentally different in generator learning. The generators are learned through $\min_\theta \mathrm{KL}(\Pi_{\alpha_t,\phi_t}(\mathbf{x},\mathbf{z})\|P_\theta(\mathbf{x},\mathbf{z}))$ [43] or $\min_\theta \mathrm{KL}(\tilde{P}_{\theta_t,\alpha_t}(\mathbf{x},\mathbf{z})\|P_\theta(\mathbf{x},\mathbf{z}))$ [42], however, generators have no access to the empirical observations which lead to biased and sub-optimal generator models.

## 3.4 Learning Inference Model via Dual-MCMC Teaching

The inference model $q_\phi(\mathbf{z}|\mathbf{x})$ serves as a key component for generator learning which will in turn facilitate the EBM training. In this paper, the inference model is learned by minimizing the KL divergences $D_q(\phi)$ as

$$\min_\phi D_q(\phi) = \min_\phi \mathrm{KL}(\tilde{Q}_{\phi_t,\theta_t}(\mathbf{x},\mathbf{z})\|Q_\phi(\mathbf{x},\mathbf{z})) + \mathrm{KL}(\tilde{P}_{\theta_t,\alpha_t}(\mathbf{x},\mathbf{z})\|\Pi_{\alpha,\phi}(\mathbf{x},\mathbf{z}))$$

$$\text{where} \quad -\frac{\partial}{\partial\phi}D_q(\phi) = \mathbb{E}_{p_{\mathrm{d}}(\mathbf{x})\mathcal{T}^{\mathbf{z}}_{\theta_t}q_{\phi_t}(\mathbf{z}|\mathbf{x})}[\frac{\partial}{\partial\phi}\log q_\phi(\mathbf{z}|\mathbf{x})] + \mathbb{E}_{\mathcal{T}^{\mathbf{x}}_{\alpha_t}p_{\theta_t}(\mathbf{x},\mathbf{z})}[\frac{\partial}{\partial\phi}\log q_\phi(\mathbf{z}|\mathbf{x})] \quad (15)$$

The revised data density $\tilde{Q}_{\phi_t,\theta_t}$ *teaches* the inference model on empirical data observations for better real data inference through the first KL term, and the revised generator density $\tilde{P}_{\theta_t,\alpha_t}$ *teaches* the inference model for better generated sample inference through the second KL term.

**Real data inference.** Optimizing the first term $\mathrm{KL}(\tilde{Q}_{\phi_t,\theta_t}(\mathbf{x},\mathbf{z})\|Q_\phi(\mathbf{x},\mathbf{z}))$ in Eqn.15 is equivalent to $\min_\phi \mathbb{E}_{p_{\mathrm{d}}(\mathbf{x})}[\mathrm{KL}(\mathcal{T}^{\mathbf{z}}_{\theta_t}q_{\phi_t}(\mathbf{z}|\mathbf{x})\|q_\phi(\mathbf{z}|\mathbf{x}))]$. Given the long-run optimality condition of the MCMC transition $\mathcal{T}^{\mathbf{z}}_{\theta_t}q_{\phi_t}(\mathbf{z}|\mathbf{x}) \to p_{\theta_t}(\mathbf{z}|\mathbf{x})$, our first term $\mathrm{KL}(\tilde{Q}_{\phi_t,\theta_t}(\mathbf{x},\mathbf{z})\|Q_\phi(\mathbf{x},\mathbf{z}))$ tends to learn $q_\phi(\mathbf{z}|\mathbf{x})$ by minimizing the $\mathbb{E}_{p_{\mathrm{d}}(\mathbf{x})}[\mathrm{KL}(p_{\theta_t}(\mathbf{z}|\mathbf{x})\|q_\phi(\mathbf{z}|\mathbf{x}))]$. The inference model is learned to match the true generator posterior $p_\theta(\mathbf{z}|\mathbf{x})$ on real observations in the current learning step. Specifically, latent samples are initialized from current $q_{\phi_t}(\mathbf{z}|\mathbf{x})$, and the generator-guided MCMC revision $\mathcal{T}^{\mathbf{z}}_{\theta_t}q_{\phi_t}(\mathbf{z}|\mathbf{x})$ is then performed to obtain the revised latent samples. The inference model is updated to amortize the MCMC and to absorb such sample revision. The MCMC revision not only drives the evolution of the latent samples, but also drives the evolution of the inference model.

**Generated sample inference.** Optimizing the second term $\mathrm{KL}(\tilde{P}_{\theta_t,\alpha_t}(\mathbf{x},\mathbf{z})\|\Pi_{\alpha,\phi}(\mathbf{x},\mathbf{z}))$ in Eqn.15 is equivalent to $\min_\phi \mathbb{E}_{\mathcal{T}^{\mathbf{x}}_{\alpha_t}p_{\theta_t}(\mathbf{x})}[\mathrm{KL}(p_{\theta_t}(\mathbf{z}|\mathbf{x})\|q_\phi(\mathbf{z}|\mathbf{x}))]$ which tends to minimizing $\mathbb{E}_{\pi_{\alpha_t}(\mathbf{x})}[\mathrm{KL}(p_{\theta_t}(\mathbf{z}|\mathbf{x})\|q_\phi(\mathbf{z}|\mathbf{x}))]$ given long-run optimality condition (i.e., $\mathcal{T}^{\mathbf{x}}_{\alpha_t}p_{\theta_t}(\mathbf{x}) \to \pi_{\alpha_t}(\mathbf{x})$). The inference model is learned to match the true generator posterior $p_\theta(\mathbf{z}|\mathbf{x})$ on generated samples from EBM in the current learning step. Noted that both the generated sample and its latent factor can be readily available where the latent factor $\mathbf{z}$ is drawn from prior distribution $p(\mathbf{z})$, which is assumed to be unit Gaussian, and the generated sample is obtained directly from generator.

## 4 Related Work

**Energy-based model.** The EBM is flexible and theoretically appealing with various approaches to learning, such as the noise-contrastive estimation (NCE) [1, 38] and the diffusion approach [8]. Most existing works learn the EBM via MLE [29, 30, 5, 6, 39], which typically involves MCMC sampling, while some advance [15, 16, 12, 17, 24] propose to amortize MCMC sampling with the generator model and learn EBM in a close-formed, variational learning scheme. Instead, [42, 43] recruit ancestral Langevin dynamics with the generator model being the initializer model. In this paper, we propose a joint framework where the generator model matches both the EBM and empirical data distribution through *dual-MCMC teaching* to better benefit the EBM sampling and learning.

**Generator model.** In recent years, the success of generator models has given rise to various ways of learning methods. Generative adversarial network (GAN) [10, 19, 27, 20] jointly trains the generator model with a discriminator, while VAE [21, 32, 36, 9] is trained with an inference model (or encoder) approximating the generator posterior. Without the inference model, [14, 31] instead utilize MCMC sampling to sample from the generator posterior. Our work differs from theirs by employing the MCMC inference based on informative initialization from the inference model, and we aim to learn the generator model to facilitate effective learning of the EBM.

## 5 Experiment

In this section, we address the following questions: (1) Can our method learn an EBM with high-quality synthesis? (2) Can both the complementary generator and inference model successfully match their MCMC-revised samples? and (3) What is the influence of the inference model and generator model? We refer to implementation details and additional experiments in Appendix.

### 5.1 Image Modelling

We first evaluate the EBM in image data modelling. Both the generator model and EBM are learned to match empirical data distribution, and if the generator model is well-trained, it can serve as an informative initializer model, making the EBM sampling easier. As a result, the EBM should be capable of generating realistic image synthesis. For evaluation, we generate images from the EBM by obtaining $x_0$ from the generator and running Langevin dynamics with $x_0$ being the initial points.

Table 1: FID and IS on CIFAR-10 and CelebA-64.

| Methods | CIFAR-10 | | CelebA-64 |
| | IS ($\uparrow$) | FID ($\downarrow$) | FID ($\downarrow$) |
| --- | --- | --- | --- |
| **Ours** | **8.55** | **9.26** | **5.15** |
| Cooperative EBM [42] | 6.55 | 33.61 | 16.65 |
| Amortized EBM [43] | 6.65 | - | - |
| Divergence Triangle [15] | 7.23 | 30.10 | 18.21 |
| No MCMC EBM [12] | - | 27.5 | - |
| Short-run EBM [29] | 6.21 | - | 23.02 |
| IGEBM [5] | 6.78 | 38.2 | - |
| ImprovedCD EBM [6] | 7.85 | 25.1 | - |
| Diffusion EBM [8] | 8.30 | 9.58 | 5.98 |
| VAEBM [39] | 8.43 | 12.19 | 5.31 |
| NCP-VAE[1] | - | 24.08 | 5.25 |
| SNGAN [27] | 8.22 | 21.7 | 6.1 |
| StyleGANv2 w/o ADA[20] | 8.99 | 9.9 | 2.32 |
| NCSN[34] | 8.87 | 25.32 | 25.30 |
| DDPM[18] | 9.46 | 3.17 | 3.93 |

Table 2: FID on CelebA-HQ-256 and LSUN-Church-64.

| Methods | CelebA-HQ-256 | LSUN-Church-64 |
| --- | --- | --- |
| **Ours** | **15.89** | **4.56** |
| Diffusion EBM [8] | - | 7.02 |
| VAEBM [39] | 20.38 | 13.51 |
| NCP-VAE[1] | 27.79 | - |
| GLOW[22] | 68.93 | 59.35 |
| PGGAN [19] | 21.7 | 6.1 |

Figure 1: Image synthesis on CIFAR-10.

We benchmark our method on standard datasets such as CIFAR-10 [23] and CelebA-64 [25], as well as challenging high-resolution CelebA-HQ-256 [19] and large-scale LSUN-Church-64 [44]. We consider the baseline models, including Divergence Triangle [15], No MCMC EBM [12], Cooperative EBM [42], and Amortized EBM [43], as well as modern advanced generative models, including other EBMs [29, 6, 8, 39, 1], GANs [27, 20] and score-based models [34, 18]. We recruit Fréchet Inception Distance (FID) and Inception Score (IS) metrics to evaluate the quality of image synthesis. Results are reported in Tab.1 and Tab.2 where our EBM shows the capability of generating realistic image synthesis and renders competitive performance even compared to GANs and score-based models.

## 5.2 MCMC Revision

The complementary generator and inference model are learned to match their MCMC-revised samples and thus can serve as informative initializers. We demonstrate that both the generator and inference model can successfully catch up with the MCMC revision. We train our model on CelebA-64 using Langevin steps $k_{\mathbf{x}} = 30$ for the MCMC revision on $\mathbf{x}$ and $k_{\mathbf{z}} = 10$ for the MCMC revision on $\mathbf{z}$.

**Generator model.** If the generator model captures different modes of the EBM, the MCMC revision on $\mathbf{x}$ should only need to search around the local mode and correct pixel-level details. To examine the generator model, we visualize the Langevin transition by drawing $\mathbf{x}_i$ for every three steps from generated samples $\mathbf{x}_0$ to MCMC-revised samples $\mathbf{x}_k$. As shown in Fig.2, only minor changes can be observed during the transition, suggesting that the generator has matched the EBM-guided MCMC revision. By measuring FID of $\mathbf{x}_0$ and $\mathbf{x}_k$, it still improves from 5.94 to 5.15, which indicates pixel-level refinements.

**Inference model.** We then show the Langevin transition on $\mathbf{z}$. For visualization, latent codes are mapped to data space via the generation network. We draw $\mathbf{z}_i$ for each step and show corresponding images in Fig.2, where the inference model also catches up with the generator-guided explaining-away MCMC inference, leading to faithful reconstruction as a result.

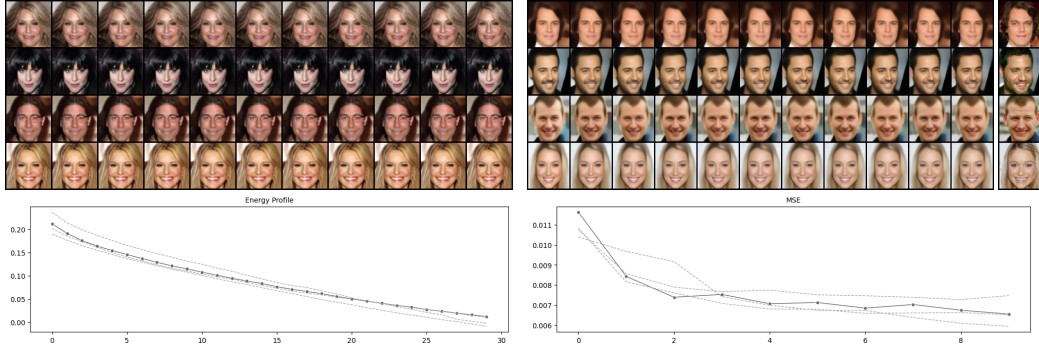

Figure 2: **Left top:** MCMC revision on $\mathbf{x}$. The leftmost images are sampled from the generator model, and the rightmost images are at the final step of the EBM-guided MCMC sampling. **Left bottom:** Energy profile over steps. **Right top:** MCMC revision on $\mathbf{z}$. The leftmost images are reconstructed by latent codes inferred from the inference model, and the rightmost images are reconstructed by latent codes at the final step of the generator-guided MCMC inference. **Right bottom:** Mean Squared Error (MSE) over steps.

## 5.3 Analysis of Inference Model

The inference model serves as an initializer model for generator learning which in turn facilitates the EBM sampling and learning. To demonstrate the benefit of the inference model, we adopt noise-initialized Langevin dynamics for generator posterior sampling and compare with the Langevin dynamics initialized by the inference model.

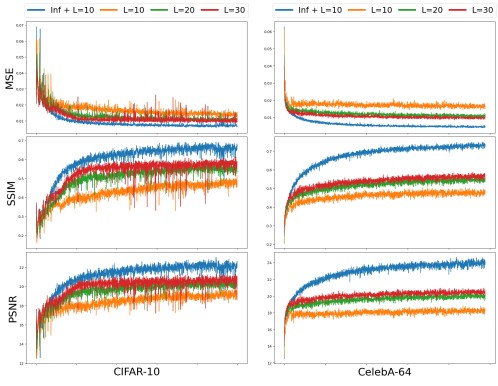

Figure 3: MSE($\downarrow$), SSIM($\uparrow$) and PSNR ($\uparrow$).

Table 3: Comparison of MSE. **Inf+L=10** denotes using Langevin dynamics initialized by inference model for $k_{\mathbf{z}} = 10$ steps.

| Methods | CIFAR-10 | CelebA-64 |
|---|---|---|
| VAE[21] | 0.0341 | 0.0438 |
| WAE[36] | 0.0291 | 0.0237 |
| RAE[9] | 0.0231 | 0.0246 |
| ABP[14] | 0.0183 | 0.0277 |
| SR-ABP[31] | 0.0262 | 0.0330 |
| Cooperative EBM[42] | 0.0271 | 0.0387 |
| Divergence Triangle[15] | 0.0237 | 0.0281 |
| Ours (Inf) | 0.0214 | 0.0227 |
| **Ours (Inf+L=10)** | **0.0072** | **0.0164** |

Specifically, we conduct noise-initialized Langevin dynamics with increasing steps from $k_{\mathbf{z}} = 10$ to $k_{\mathbf{z}} = 30$, and compare with the Langevin dynamics using only $k_{\mathbf{z}} = 10$ steps but is initialized by the inference model. We recruit MSE, Peak Signal-to-Noise Ratio (PSNR), and Structural SIMilarity (SSIM) to measure the inference accuracy of reconstruction and present the results in Fig.3. As the Langevin steps increase, the inference becomes more accurate (lower MSE, higher PSNR, and SSIM), however, it is still less accurate than the proposed method (L=30 vs. Inf+L=10). This result highlights the inference model in our framework. We then compare with other models that also characterize an inferential mechanism, such as VAE [21], Wasserstein auto-encoders (WAE) [36], RAE [9], Alternating Back-propagation (ABP) [14], and Short-run ABP (SR-ABP) [31]. As shown in Tab.3, our model can render superior performance with faithful reconstruction.

## 5.4  Analysis of Generator Model

With the generator model being the initializer for EBM sampling, exploring the energy landscape should become easier by first traversing the low-dimensional latent space. We intend to examine if our generator model can deliver smooth interpolation on the latent space, thus making a smooth transition in the data space. We employ linear interpolation among latent space, i.e., $\tilde{\mathbf{z}} = (1 - \alpha) \cdot \mathbf{z}_1 + \alpha \cdot \mathbf{z}_2$, and consider two scenarios, such as the image synthesis and image reconstruction. As shown in Fig.4, our generator model is capable of smooth interpolation for both scenarios, which suggests its effectiveness in exploring the energy landscape.

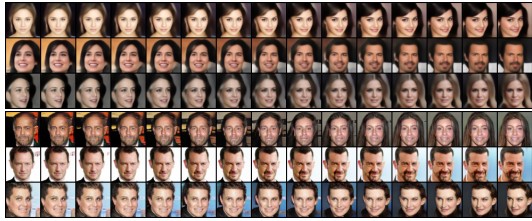

Figure 4: Linear interpolation on latent space. The *top* and *bottom* three rows indicate image generation and reconstruction, respectively.

## 6  Ablation Studies

**MCMC steps of $\mathcal{T}_\theta^{\mathbf{z}}$.** We analyze the impact of the inference accuracy in our framework by increasing the Langevin steps of $\mathcal{T}_\theta^{\mathbf{z}} q_\phi(\mathbf{z}|\mathbf{x})$. With an inference model initializing the MCMC posterior sampling, further increasing the MCMC steps should deliver more accurate inference and thus benefit the generator and EBM for better performance. Thus, we compute the FID, MSE, and wall-clock training time (seconds / per iteration) in Tab.4. It can be seen that increasing MCMC steps from 10 to 30 indeed slightly improves the generation quality and inference accuracy but requires more training time. We thus report the result of *Inf+L=10* in Tab.1 and Tab.3.

**MCMC steps of $\mathcal{T}_\alpha^{\mathbf{x}}$.** Then, we discuss the impact of the Langevin steps of $\mathcal{T}_\alpha^{\mathbf{x}}$. Increasing the MCMC steps of $\mathcal{T}_{\mathbf{x}}$ should explore the energy landscape more effectively and render better performance in the generation. In Tab.5, starting MCMC steps from 10 to 30, our model exhibits largely improved performance in generation quality but only minor improvement even when we use $L = 50$ steps. Thus, we report $L = 30$ steps in Tab.1.

Table 4: Increasing MCMC steps of $\mathcal{T}_\theta^{\mathbf{z}}$.

|  | L=10 | L=30 | **Inf+L=10** | Inf+L=30 |
|---|---|---|---|---|
| FID | 17.32 | 14.51 | **9.26** | 9.18 |
| MSE | 0.0214 | 0.0164 | 0.0072 | 0.0068 |
| Time (s) | 1.576 | 2.034 | 1.594 | 2.112 |

Table 5: Increasing MCMC steps of $\mathcal{T}_\alpha^{\mathbf{x}}$.

|  | L=10 | L=20 | **L=30** | L=50 |
|---|---|---|---|---|
| FID | 14.78 | 11.51 | **9.26** | 9.07 |
| Time (s) | 0.861 | 1.241 | 1.594 | 2.454 |

## 7  Conclusion

We present a joint learning scheme that can effectively learn the EBM by interweaving the maximum likelihood learning of the EBM, generator, and inference model through dual-MCMC teaching. The generator and inference model are learned to initialize MCMC sampling of EBM and generator posterior, respectively, while these EBM-guided MCMC sampling and generator-guided MCMC inference, in turn, serve as two MCMC revision processes that are capable of *teaching* the generator and inference model. This work may share the limitation with other MCMC-based methods in terms of the computational cost, but we expect to impact the active research of learning the EBMs.

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
