# Supplementary Material for Learning Energy-based Model via Dual-MCMC Teaching

**Jiali Cui, Tian Han**
Department of Computer Science, Stevens Institute of Technology
{jcui7,than6}@stevens.edu

## 1  Addtional Experiment

We show additional image synthesis in Fig.2. Images are sampled from EBM with the initial point generated by the generator.

### 1.1  Parameter Efficiency

To further illustrate the effectiveness of our method, we follow baseline models [5, 8] and recruit simple convolution networks for the EBM, generator, and inference models. We train our model with such a simple structure on CIFAR-10 and report the results in Tab.1. It can be seen that even though using simple network structures, the proposed method can still generate realistic image synthesis.

For reported numbers in main text, we adopt the network structure that contains *Residue Blocks* (see implementation details in Tab.5). Such a network structure is commonly used in EBM works [1, 2, 7, 6]. To shed further light on our method, we increase the hidden features (denoted as **nef**) and report the result in Tab.2. We observe that using small **nef**=256 still shows strong performance, while increasing from **nef**=512 to **nef**=1024 only exhibits minor improvement. This highlights the effectiveness endowed with the proposed learning scheme.

Table 1: FID for simple network structure.

|  | Cooperative EBM [8] | Divergence Triangle [5] | No MCMC EBM [4] | Ours |
|---|---|---|---|---|
| FID | 33.61 | 30.10 | 27.50 | 19.35 |

Table 2: FID for increasing **nef**.

|  | **nef**=256 | **nef**=512 | **nef**=1024 |
|---|---|---|---|
| FID | 11.19 | **9.26** | 8.45 |

### 1.2  Out-of-Distribution Detection

We evaluate our EBM in out-of-distribution (OOD) detection task. If the EBM is well-learned, it can be viewed as a generative discriminator and is able to distinguish the in-distribution data with a lower energy value and out-of-distribution data by assigning a higher energy value. We follow the protocol [7] and train our EBM on CIFAR-10. We test with multiple OOD data and compute the energy value as the decision function. Tab.3 shows the performance evaluated by the AUROC score, where our EBM performs well compared to other unsupervised learning methods and can be competitive even compared with the supervised (label available) methods.

Table 3: AUROC ($\uparrow$) for OOD detection.

|  | SVHN | CIFAR-100 | CelebA |
|---|---|---|---|
| **Unsupervised Method** |  |  |  |
| Ours | **0.94** | **0.64** | **0.85** |
| Divergence Triangle [5] | 0.68 | - | 0.56 |
| No MCMC EBM [4] | 0.83 | 0.73 | 0.33 |
| IGEBM [1] | 0.63 | 0.50 | 0.70 |
| ImprovedCD EBM [2] | 0.91 | 0.83 | - |
| VAEBM [7] | 0.83 | 0.62 | 0.77 |
| **Supervised Method** |  |  |  |
| JEM [3] | 0.67 | 0.67 | 0.75 |
| HDGE | 0.96 | 0.91 | 0.80 |
| OOD EBM | 0.91 | 0.87 | 0.78 |
| OOD EBM (fine-tuned) | 0.99 | 0.94 | 1.00 |

37th Conference on Neural Information Processing Systems (NeurIPS 2023).

## 1.3 Image Inpainting

We then test our model for the task of image inpainting. We show that our method is capable of recovering occluded images by progressively involving two MCMC revision processes. Specifically, we consider the increasingly challenging experiment settings: (1) M20, M30, M40 are denoted for center block of size 20x20, 30x30, 40x40, (2) R20, R30, R40 are denoted for multiple blocks that cover 20%, 30%, 40% pixels of the original images. For recovery, we take occluded images as input for the inference model and feed inferred latent codes through the generator model for recovery. The performance of recovery should become better after the MCMC revision. As shown in Fig.1, our model successfully recovers occluded images with MCMC revision processes.

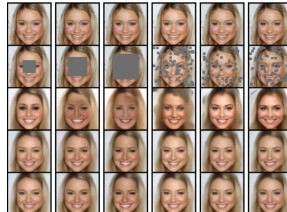

| PSNR / SSIM | M20 | M30 | M40 |
|---|---|---|---|
| Inf+Gen | 21.035 / 0.671 | 18.375 / 0.568 | 16.484 / 0.487 |
| Inf+$\mathcal{T}_\theta^\mathbf{z}$+Gen | 24.976 (↑) / 0.781 (↑) | 23.085 (↑) / 0.747 (↑) | 19.733 (↑) / 0.660 (↑) |
| Inf+$\mathcal{T}_\theta^\mathbf{z}$+Gen+$\mathcal{T}_\alpha^\mathbf{x}$ | 25.132 (↑) / 0.797 (↑) | 23.276 (↑) / 0.763 (↑) | 19.959 (↑) / 0.679 (↑) |
| PSNR / SSIM | R20 | R30 | R40 |
| Inf+Gen | 18.174 / 0.558 | 17.092 / 0.507 | 16.348 / 0.472 |
| Inf+$\mathcal{T}_\theta^\mathbf{z}$+Gen | 25.273 (↑) / 0.779 (↑) | 25.108 (↑) / 0.771 (↑) | 24.923 (↑) / 0.769 (↑) |
| Inf+$\mathcal{T}_\theta^\mathbf{z}$+Gen+$\mathcal{T}_\alpha^\mathbf{x}$ | 25.666 (↑) / 0.793 (↑) | 25.409 (↑) / 0.788 (↑) | 25.171 (↑) / 0.781 (↑) |

Figure 1: Visualization of image completion. From *top* to *bottom* row: test image, occluded image, recovery image via (i) Inf+Gen, (ii) Inf+$\mathcal{T}_\theta^\mathbf{z}$+Gen, (iii) Inf+$\mathcal{T}_\theta^\mathbf{z}$+Gen+$\mathcal{T}_\alpha^\mathbf{x}$. From *left* to *right* column: experiments settings of M20, M30, M40, R20, R30, R40.

# 2 Theoretical Derivations

## 2.1 Preliminary

**Learning generator model:** Recall that the generator model is specified as $p_\theta(\mathbf{x}, \mathbf{z})$ and can be learned by maximizing its log-likelihood $L_p(\theta) = \log p_\theta(\mathbf{x})$. The learning gradient is based on the simple identity: $\frac{\partial}{\partial\theta}\log p_\theta(\mathbf{x}) = \int \frac{\partial}{\partial\theta}\log p_\theta(\mathbf{x},\mathbf{z})\frac{p_\theta(\mathbf{x},\mathbf{z})}{p_\theta(\mathbf{x})}d\mathbf{z} = \mathbb{E}_{p_\theta(\mathbf{z}|\mathbf{x})}[\frac{\partial}{\partial\theta}\log p_\theta(\mathbf{x},\mathbf{z})]$.

**Learning energy-based model:** For learning the EBM $\pi_\alpha(\mathbf{x})$, the gradient is computed by maximizing its log-likelihood as $\frac{\partial}{\partial\alpha}\log \pi_\alpha(\mathbf{x}) = \frac{\partial}{\partial\alpha}[f_\alpha(\mathbf{x}) - \log \mathrm{Z}(\alpha)]$, where $\frac{\partial}{\partial\alpha}\log \mathrm{Z}(\alpha) = \frac{1}{\mathrm{Z}(\alpha)}\int \frac{\partial}{\partial\alpha}\exp[f_\alpha(\mathbf{x})]d\mathbf{x} = \int \pi_\alpha(\mathbf{x})\frac{\partial}{\partial\alpha}f_\alpha(\mathbf{x})d\mathbf{x} = \mathbb{E}_{\pi_\alpha(\mathbf{x})}[\frac{\partial}{\partial\alpha}f_\alpha(\mathbf{x})]$.

## 2.2 Methodology

**Joint desity & Marginal density.** Given the KL divergence between two arbitrary joint densities, i.e., $\mathrm{KL}(p(\mathbf{x},\mathbf{z})\|q(\mathbf{x},\mathbf{z}))$, one could obtain the following identity,

$$
\begin{aligned}
\mathrm{KL}(p(\mathbf{x},\mathbf{z})\|q(\mathbf{x},\mathbf{z})) &= \int\int p(\mathbf{x},\mathbf{z})\log\frac{p(\mathbf{x},\mathbf{z})}{q(\mathbf{x},\mathbf{z})}d\mathbf{x}d\mathbf{z} \\
&= \int p(\mathbf{x})\log\frac{p(\mathbf{x})}{q(\mathbf{x})}d\mathbf{x} + \int\int p(\mathbf{x},\mathbf{z})\log\frac{p(\mathbf{z}|\mathbf{x})}{q(\mathbf{z}|\mathbf{x})}d\mathbf{x}d\mathbf{z} \\
&= \mathrm{KL}(p(\mathbf{x})\|q(\mathbf{x})) + \mathbb{E}_{p(\mathbf{x})}[\mathrm{KL}(p(\mathbf{z}|\mathbf{x})\|q(\mathbf{z}|\mathbf{x}))]
\end{aligned}
\tag{1}
$$

which derives the marginal version of KL divergences of Eqn.8, Eqn.11, and Eqn.13 in the main text.

**MLE perturbation for EBM.** The EBM is learned through the minimization of joint KL divergences as $\min_\alpha \mathrm{KL}(\tilde{Q}_{\phi_t,\theta_t}(\mathbf{x},\mathbf{z})\|\Pi_{\alpha,\phi}(\mathbf{x},\mathbf{z})) - \mathrm{KL}(\tilde{P}_{\theta_t,\alpha_t}(\mathbf{x},\mathbf{z})\|\Pi_{\alpha,\phi}(\mathbf{x},\mathbf{z}))$. With Eqn.1, we could have

$$
\begin{aligned}
&\min_\alpha \mathrm{KL}(\tilde{Q}_{\phi_t,\theta_t}(\mathbf{x},\mathbf{z})\|\Pi_{\alpha,\phi}(\mathbf{x},\mathbf{z})) - \mathrm{KL}(\tilde{P}_{\theta_t,\alpha_t}(\mathbf{x},\mathbf{z})\|\Pi_{\alpha,\phi}(\mathbf{x},\mathbf{z})) \\
&= \min_\alpha \mathrm{KL}(p_\mathrm{d}(\mathbf{x})\|\pi_\alpha(\mathbf{x})) + C_1 - \mathrm{KL}(\mathcal{T}_{\alpha_t}^\mathbf{x} p_{\theta_t}(\mathbf{x})\|\pi_\alpha(\mathbf{x})) - C_2
\end{aligned}
$$

where $C_1$ $(= \mathrm{KL}(\mathcal{T}_{\theta_t}^\mathbf{z}q_{\phi_t}(\mathbf{z}|\mathbf{x})\|q_\phi(\mathbf{z}|\mathbf{x})))$ and $C_2$ $(= \mathrm{KL}(p_{\theta_t}(\mathbf{z}|\mathbf{x})\|q_\phi(\mathbf{z}|\mathbf{x})))$ are constant irrelevant to learning parameters. This is the marginal version of Eqn.8 shown in the main text.

## 2.3 Learning Algorithm

Our probabilistic framework consists of the EBM $\pi_\alpha$, generator model $p_\theta$, and inference model $q_\phi$. Three models are trained in an alternative and iterative manner based on the current model parameters. Specifically, recall that the joint KL divergences between *revised densities* $\tilde{Q}_{\phi,\theta}(\mathbf{x}, \mathbf{z})$, $\tilde{P}_{\theta,\alpha}(\mathbf{x}, \mathbf{z})$ and model densities give the gradient:

$$-\frac{\partial}{\partial \alpha} D_\pi(\alpha) = \mathbb{E}_{p_\mathrm{d}(\mathbf{x})}\big[\frac{\partial}{\partial \alpha} f_\alpha(\mathbf{x})\big] - \mathbb{E}_{\mathcal{T}_{\alpha_t}^{\mathbf{x}} p_{\theta_t}(\mathbf{x})}\big[\frac{\partial}{\partial \alpha} f_\alpha(\mathbf{x})\big] \tag{2}$$

$$-\frac{\partial}{\partial \theta} D_p(\theta) = \mathbb{E}_{p_\mathrm{d}(\mathbf{x})\mathcal{T}_{\theta_t}^{\mathbf{z}} q_{\phi_t}(\mathbf{z}|\mathbf{x})}\big[\frac{\partial}{\partial \theta} \log p_\theta(\mathbf{x}, \mathbf{z})\big] + \mathbb{E}_{\mathcal{T}_{\alpha_t}^{\mathbf{x}} p_{\theta_t}(\mathbf{x}|\mathbf{z})p(\mathbf{z})}\big[\frac{\partial}{\partial \theta} \log p_\theta(\mathbf{x}, \mathbf{z})\big] \tag{3}$$

$$-\frac{\partial}{\partial \phi} D_q(\phi) = \mathbb{E}_{p_\mathrm{d}(\mathbf{x})\mathcal{T}_{\theta_t}^{\mathbf{z}} q_{\phi_t}(\mathbf{z}|\mathbf{x})}\big[\frac{\partial}{\partial \phi} \log q_\phi(\mathbf{z}|\mathbf{x})\big] + \mathbb{E}_{\mathcal{T}_{\alpha_t}^{\mathbf{x}} p_{\theta_t}(\mathbf{x},\mathbf{z})}\big[\frac{\partial}{\partial \phi} \log q_\phi(\mathbf{z}|\mathbf{x})\big] \tag{4}$$

Each model can then be updated via stochastic gradient ascent with such gradient.

Computing the above gradient needs the MCMC sampling and the MCMC inference as two MCMC revision processes. We adopt the Langevin dynamics that iterates as

$$\mathbf{x}_{\tau+1} = \mathbf{x}_\tau + s\frac{\partial}{\partial \mathbf{x}_\tau} \log \pi_\alpha(\mathbf{x}_\tau) + \sqrt{2s}U_\tau \quad \text{where} \quad \mathbf{x}_0 \sim p_\theta(\mathbf{x}, \mathbf{z}) \quad \text{and} \quad \mathbf{z} \sim \mathcal{N}(0, I_d) \tag{5}$$

$$\mathbf{z}_{\tau+1} = \mathbf{z}_\tau + s\frac{\partial}{\partial \mathbf{z}_\tau} \log p_\theta(\mathbf{z}_\tau|\mathbf{x}) + \sqrt{2s}U_\tau \quad \text{where} \quad \mathbf{z}_0 \sim q_\phi(\mathbf{z}|\mathbf{x}) \quad \text{and} \quad \mathbf{x} \sim p_\mathrm{d}(\mathbf{x}) \tag{6}$$

Compared to Eqn.3 and Eqn.6 in the main text, Eqn.5 and Eqn.6 start with initial points initialized by the generator and inference model, respectively. The final $\mathbf{x}_\tau$ and $\mathbf{z}_\tau$ are sampled through the guidance of EBM and generator model, and they serve as two MCMC-revised samples that teach the initializer models.

We present the learning algorithm in Alg.1.

---

**Algorithm 1** Learning EBM, generator and inference model and via *dual-MCMC teaching*

---

**Require:**
    Batch size $B$. Training images $\{\mathbf{x}_i\}_{i=1}^B$.
    Total learning iterations $T$. Current learning iterations $t$.
    Network parameters $\alpha$, $\theta$, $\phi$. Fixed parameters $\alpha_t$, $\theta_t$, $\phi_t$.
    Let $t \leftarrow 0$.
    **repeat**
        **Training samples:** Let $\mathbf{x} = \{\mathbf{x}_i\}_{i=1}^B$.
        **Prior latent:** Let $\mathbf{z} = \{\mathbf{z}_i\}_{i=1}^B$, where $\{\mathbf{z}_i\}_{i=1}^B \sim \mathcal{N}(0, I_d)$.
        **MCMC Sampling:** Sample $\hat{\mathbf{x}}$ from generator model $\theta_t$ using $\mathbf{z}$. Sample $\tilde{\mathbf{x}}$ using Eqn.5 with $\alpha_t$ and $\hat{\mathbf{x}}$ being initial points.
        **MCMC Inference:** Sample $\hat{\mathbf{z}}$ from inference model $\phi_t$ using $\mathbf{x}$. Sample $\tilde{\mathbf{z}}$ using Eqn.6 with $\theta_t$ and $\hat{\mathbf{z}}$ being initial points.
        **Learn $\pi_\alpha$:** Update $\alpha$ using Eqn.2 with $\mathbf{x}$ and $\tilde{\mathbf{x}}$.
        **Learn $p_\theta$:** Update $\theta$ using Eqn.3 with $\mathbf{x}$, $\tilde{\mathbf{z}}$, $\tilde{\mathbf{x}}$, and $\mathbf{z}$.
        **Learn $q_\phi$:** Update $\phi$ using Eqn.4 with $\mathbf{x}$, $\tilde{\mathbf{z}}$, $\tilde{\mathbf{x}}$, and $\mathbf{z}$.
        Let $t \leftarrow t + 1$.
    **until** $t = T$

---

## 2.4 Computational and Memory Cost

Our learning algorithm belongs to MCMC-based methods and can incur computational overhead due to its iterative nature compared to variational-based or adversarial methods. We provide further analysis by computing the wall-clock training time and parameter complexity for our related work Divergence Triangle [5] (variational and adversarial-based joint training without MCMC) and our model (see Tab.4), where the proposed work requires more training time but can also render significantly better performance. Regarding memory cost, it's important to note that we didn't observe further improvement by just increasing parameter complexity (see Sec.1.1). This emphasizes the effectiveness provided by our learning algorithm.

Table 4: Comparison between Divergence Triangle and our model for sample quality, wall-clock training time (seconds / per-iteration), network parameters (denoted as #). Our method[1] uses the same network as Divergence Triangle, while method[2] utilizes more complex residual network structures.

| | Divergence Triangle[5] | Ours[1] | Ours[2] |
|---|---|---|---|
| FID | 30.10 | 19.35 | **9.26** |
| Time (s) | 0.092 | 0.201 | 1.594 |
| # Generator | 8M | 8M | 16M |
| # Inference | 5M | 5M | 15M |
| # EBM | 2M | 2M | 16M |
| Langevin Steps on $\mathbf{x}$ | 0 | 30 | 30 |
| Langevin Steps on $\mathbf{z}$ | 0 | 10 | 10 |

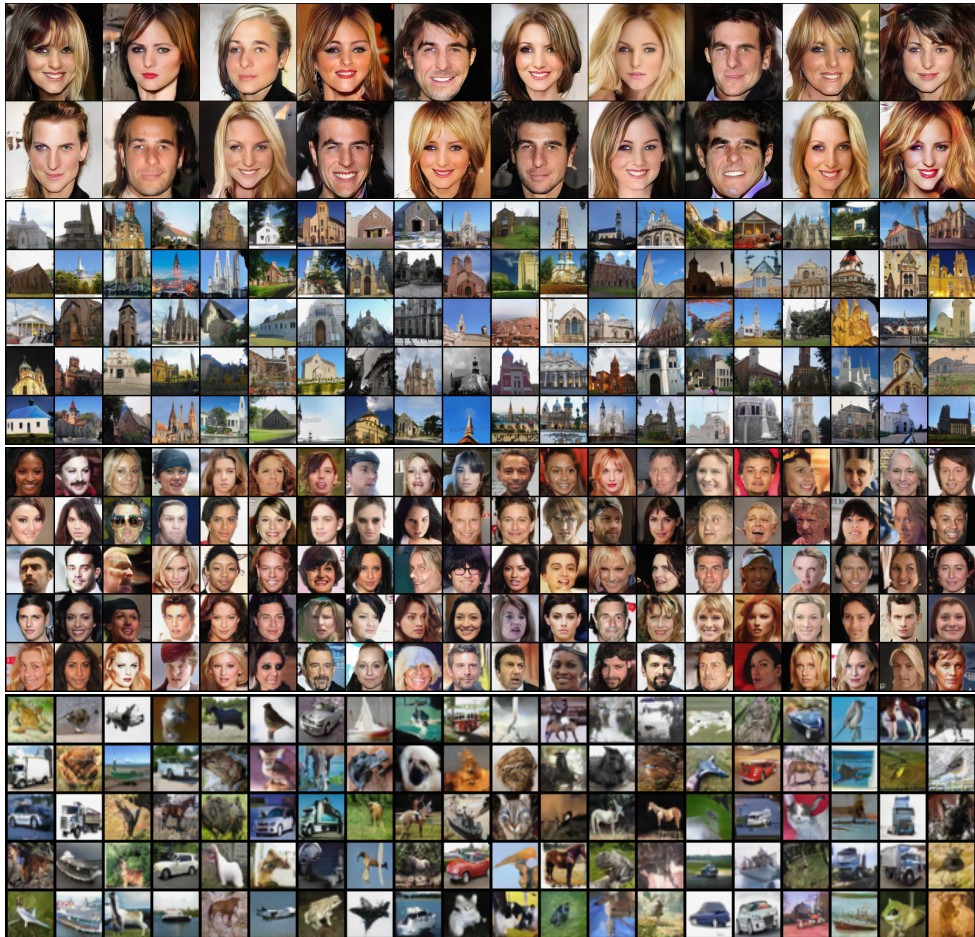

Figure 2: Additional results for image synthesis. From top to bottom: CelebA-HQ-256, LSUN-Church-64, CelebA-64, CIFAR-10.

## 3   Experiment Detail

We compute FID scores with 30,000 generated images for CelebA-HQ-256 and 50,000 generated images for other data. All training images are resized and scaled to [-1, 1]. All experiment results run on one NVIDIA A100 GPU (40-GB).

The network structures of each model are shown in Tab.5.

Table 5: Network structures on CIFAR-10. We denote the operation of convolution and transposed convolution as **Conv** (input channel, output channel, k=3, s=1, p=1) and **ConvT** (input channel, output channel, k=3, s=1, p=1), where k is the kernel size, s is the stride number, and p is padding value. We conduct Upsample and Downsample via *interpolate* and *avg_pool2d* operations.

| **Generator Block** (in_ch, out_ch, upsample) |
| --- |
| Input: x |
| BatchNorm(in_ch), ReLU |
| Upsample(factor=2) if upsample |
| ConvT(in_ch, out_ch), BatchNorm(out_ch), ReLU |
| ConvT(out_ch, out_ch) |
| output: h |
| Input: x |
| Upsample(factor=2), ConvT(in_ch, out_ch) if upsample |
| output: y |
| output: h + y |
| **Generator Network** (z_dim, ngf) |
| Input: z |
| Linear(z_dim, 4∗4∗ngf) |
| Generator Block (ngf, ngf, upsample=True) |
| Generator Block (ngf, ngf, upsample=True) |
| Generator Block (ngf, ngf, upsample=True) |
| BatchNorm(ngf), ReLU |
| ConvT(ngf, 3), Tanh |
| output: h |
| **Hyper-parameters** |
| **ngf**=512, **nef**=512, **nif**=128, **z_dim**=128, **batch size**=100 |
| **learning rate:** (Gen) 3e-4, (EBM) 1e-4, (Inf) 1e-4 |

| **EBM Block** (in_ch, out_ch, downsample, head) |
| --- |
| Input: x |
| ReLU if head |
| Conv(in_ch, out_ch), ReLU, Conv(out_ch, out_ch) |
| Downsample(factor=2) if downsample |
| output: h |
| Input: x |
| Downsample(factor=2) if head |
| ConvT(in_ch, out_ch) if downsample |
| Downsample(factor=2) if downsample and not head |
| output: y |
| output: h + y |
| **EBM Network** (nc=3, nef) |
| Input: x |
| EBM Block(nc, nef, downsample=True, head=True) |
| EBM Block (nef, nef, downsample=True) |
| EBM Block (nef, nef, downsample=False) |
| EBM Block (nef, nef, downsample=False) |
| ReLU, Downsample(factor=8), Linear(nef, 1) |
| output: h |
| **Inference Network** (z_dim, nif) |
| Input: x |
| ConvT(3, nif), ReLU |
| ConvT(nif, nif∗2, 4, 2, 1), ReLU |
| ConvT(nif∗2, nif∗4, 4, 2, 1), ReLU |
| ConvT(nif∗4, nif∗8, 4, 2, 1), ReLU |
| output: ConvT(nif∗8, z_dim, 4, 1, 0), ConvT(nif∗4, z_dim, 4, 1, 0) |