# OpenReview forum: "Learning Energy-based Model via Dual-MCMC Teaching"
_NeurIPS.cc/2023/Conference — NeurIPS 2023 poster_

### Official Review · Reviewer_XcPg · 2023-07-02

**Soundness:** 3 good
**Presentation:** 4 excellent
**Contribution:** 3 good
**Rating:** 6
**Confidence:** 5

**Summary:**

This paper proposes a novel method for training and inference of energy-based model (EBM). The author claim that although the use of generator as an initializer model may improve MCMC sampling, training unbiased generator is an open problem . Thus they propose a joint learning framework in which generator is learned by MLE via posterior sampling. While EBMs have a number of appealing properties they are difficult to inference. Therefore learning an unbiased generator for faster MCMC sampling from the EBM distribution is a hot topic.

**Strengths:**

The paper is clearly written and well presented. The potential is impact of the proposed dual-MCMC teaching is considerable. The problem arised in the paper is actual and mostly uncovered by the previous research. The paper provides a number of novel ideas which could encorage the future research in this direction. The experiments are very clear and solid including fair comparison with concurent works. Remarkably, the numerical results for the proposed method are spectacular.


**Weaknesses:**

1. The section 2 is clear-written and self-contained but it is large and has little connection to the methodology proposed in the paper. As the authors reforlulate each component described in the section 2 it might be helpful to better explain the benefits of the choice for proposed method.

2. The objective proposed for learning EBM includes maximization of Kulback-Leibler divergence term (equation 7) which is stated to perform as a self-critic for EBM. The grounding beneath this “​​self-adversarial learning” is not very confident. Either literature referenced by the authors lack the verification or some formal reasoning behind this objective. Compared to the objectives for generator and inference model this one doesn’t pose an upper bound for MLE objective.

3. In the Figure 2 in the main text you provide an energy profile after performing $30$ steps (as during training) of MCMC revision on $x$. FID improves from $5.94$ on $x_0$ to $5.15$ on $x_{30}$, still the energy profile is not convergent. It would be more convincing if you provide a result for long-run MCMC (until energy profile convergence) revision for learned EBM to verify your EBM.

4. Please include the training details, including optimization hyperparameters, step sizes for Langevin dynamics, etc.

Typos:
1. In the line 156 there is a typo for the marginal distribution of x, but inside the brackets you use a proper notation.
2. In the line 214 “second” is used instead of “first”.


**Questions:**

1. It was noticed [1] that for EBM distribution with density $\pi(x)$ and generator $p_{\theta} = p_{\theta}(x|z)p(z)$ sampling latent variable from $p(z)\pi(g_{\theta}(z))$ and pushing it by $g_{\theta}$ to the ambient space is equivalent to sampling from $\pi(x)$. Why do you propose to perform EBM inference in the ambient space instead of latent space? As traversing high-dimensional space with MCMC sampling is a hard problem. Please elaborate on why don’t you use an advantage of sampling from EBM in the latent space of the generator model.

2. The paper massively adopts the method proposed in [2], nevertheless considering results on CIFAR-10 dataset FID drops from $30.10$ for [2] to $9.26$ for the current paper. What is a major reason for such a dramatic improvement? In the Appendix you provide a quality for a "simple network" which seems to share architecture with paper [2]. It would be helpful for understanding if you provide a comparison of model complexity and/or time, GPU consumption in addition to image quality comparison.

[1] Che, Tong, et al. "Your gan is secretly an energy-based model and you should use discriminator driven latent sampling." Advances in Neural Information Processing Systems 33 (2020): 12275-12287.

[2] Han, Tian, et al. "Divergence triangle for joint training of generator model, energy-based model, and inferential model." Proceedings of the IEEE/CVF Conference on Computer Vision and Pattern Recognition. 2019.

**Limitations:**

The limitations of the method are obscured. The authors only mention that “this work may share the limitation with other MCMC-based methods in terms of the computational cost” which is a bit vague.

---

> ### Author Rebuttal · Authors · 2023-08-08
>
> Thanks for your encouraging feedback and the correction of typos. We shall follow the suggestion and fix the typos in the revision. In the **General Response**, we provide clarifications for concerns regarding the motivation of our method, connection to Divergence Triangle, and computational cost  (see also the attached **PDF**).
>
> ---
>
> **Q1. Self-adversarial Interpretation of EBM Learning.**
>
> We learn our EBM by minimizing $\min_\alpha KL(p_{\rm data}(x)\| \pi_\alpha(x)) - KL(T^x_{\alpha_t} p_{\theta_t}(x)\|\pi_\alpha(x))$ (Eqn. 8), which actually lower bounds the traditional MLE objective. Such "$KL -KL$" objective has the self-adversarial interpretation, as the energy value for observed examples is learned to decrease, whereas it increases for model samples. Therefore, the model is learned to criticize its own samples at current iteration. On the other hand, the model sample along the MCMC iteration process (Eqn. 3) is adjusted through lowering its sample energy, which can be viewed to fool the EBM itself at the current iteration.
>
> **Q2. Long-run Langevin Trajectory.**
>
> We conduct the long-run experiment and show the result in the attached **PDF**, where we run MCMC revision on $x$ for 3000 steps (much longer than 30 training steps). We observe only minor changes during the process, which indicates that the generator model could successfully match the long-run MCMC samples from EBM and approach the target distribution $\pi_\alpha(x)$.
>
> **Q3. Training Hyper-parameters.**
>
> We shall include training details in the Appendix and open-source our implementation along with checkpoints.
>
> **Q4. Traversing Latent Space.**
>
> Thank you for providing such an insightful suggestion. As stated in [1], the data distribution $p_{\rm data}(x)$ and generator distribution $p_\theta(x)$ need to have the same support to avoid the mode-dropping phenomenon. In other words, in the case when the generator model fails to cover some of the modes in data distribution, the latent sampling from such an induced model will also miss these modes. Addressing such an issue would require an additional treatment (Corollary 1 in [1]). Our explicit EBM, on the other hand, allows a further MCMC revision, so even when the generator model only partially covers the data distribution, such MCMC revision would still effectively guide the samples toward the data manifold.
>
>
>
> Comparison of sample quality, wall-clock training time (seconds / per-iteration).
>
> |                | Traversing on $\mathbf{z}$ | Traversing on $\mathbf{x}$ |
> | :------------: | :------------------------: | :------------------------: |
> |      FID       |           40.35            |            9.26            |
> |    Time (s)    |           1.742            |           1.594            |
> | Langevin Steps |             30             |             30             |
>
>
>
> Empirically, we conducted an initial experiment where we adapted our method by traversing the induced latent space for EBM sampling, rather than directly performing MCMC on the data space. The results can be found in the table below, where we observe a weaker performance of the adapted model. Additionally, such a sampling method would increase the training time as it requires back-propagation of the gradient through the generation network. For comparison, we use the same network structure and other hyper-parameters (e.g., Langevin steps, etc.),  while we shall explore its potential in greater depth in our future study. Thanks again for such an insightful comment.
>
> **Q5. Discussion of Limitation.**
>
> The details of our computational cost can be found in **General Response**. We shall add a thorough discussion of the limitation, including our computational cost and the point (provided by **Reviewer** **uj7J**, Q1), in the revision.
>
> ---
>
> [1] Che, Tong, Ruixiang Zhang, Jascha Sohl-Dickstein, Hugo Larochelle, Liam Paull, Yuan Cao, and Yoshua Bengio. Your gan is secretly an energy-based model and you should use discriminator driven latent sampling. Advances in Neural Information Processing Systems, 33, 2020.

---

> > ### Comment · Reviewer_XcPg · 2023-08-19
> >
> > Thanks for your response and fruitful discussion, especially towards the relations of your method with the local-global MCMC approach. In general I enjoyed reading the paper and continue to support its acceptance.

---

> > > ### Author Response · Authors · 2023-08-19
> > >
> > > Thanks for your insightful comments, and we are happy to hear that you like our discussion with the local-global MCMC approach.
> > > Please feel free to let us know if you have any questions, we are willing to provide further responses.

---

### Official Review · Reviewer_iX9Y · 2023-07-03

**Soundness:** 3 good
**Presentation:** 3 good
**Contribution:** 3 good
**Rating:** 5
**Confidence:** 4

**Summary:**

This paper investigates the problem of Maximum Likelihood (ML) estimation for Energy-Based Models (EBM). Building on previous research, the authors propose a novel approach that involves learning a surrogate generative model to initialize the costly Langevin steps. The authors chose a latent-based generative model thus requiring additional Markov Chain Monte Carlo (MCMC) sampling for estimation. This new MCMC procedure is amortized using a simpler third generative model, referred to as the inference model. This work provides a procedure to learn the three models jointly. Through extensive experimental evaluations, the authors demonstrate the effectiveness of their proposed framework. They showcase the benefits of incorporating the additional models and highlight the superiority of their combined algorithm over other existing methods. The results substantiate the value of joint learning, showcasing improved accuracy.

**Strengths:**

- Experiments are well detailled.
- Experiments on Out-of-Distribution Detection (OOD) in App 1.2 allow to evaluate the EBM beyond generating samples.

**Weaknesses:**

- The motivation to learn the generator with observed samples is unclear.
- The improved accuracy of this method is not balanced with the computational and memory costs generated.
- The quality of the density estimation task is barely considered.

**Questions:**

- One of the main justifications of your learning procedure is to learn the generator using training data. Can you explain why not doing this (like in cooperative learning [1,2]) would introduce a bias ? This claim was made multiple times in the main paper.
- As a justification of your revisited densities (or in L129), you say that Langevin steps could transform something unimodal into something multimodal which is not the case (see for instance [3]) as shown in Fig 2 of Sec 5.2. Is there another justification for those Langevin steps ?
- The ablation studies (Sec 5.2 and 5.3) highlight the usefulness of the inference model with surprisingly low computational overhead. What about the memory overhead ?

[1] Jianwen Xie, Yang Lu, Ruiqi Gao, Song-Chun Zhu, and Ying Nian Wu. Cooperative training of descriptor and generator networks. IEEE transactions on pattern analysis and machine intelligence, 42(1):27–45, 2018.

[2] Jianwen Xie, Zilong Zheng, and Ping Li. Learning energy-based model with variational auto-encoder as amortized sampler. In Proceedings of the AAAI Conference on Artificial Intelligence, volume 35, pages 10441–10451, 2021

[3] Véronique Gayrard, Anton Bovier, Markus Klein, Metastability in reversible diffusion processes II: precise asymptotics for small eigenvalues. J. Eur. Math. Soc. 7 (2005), no. 1, pp. 69–99

**Limitations:**

Not applicable.

---

> ### Author Rebuttal · Authors · 2023-08-08
>
> Thanks for your detailed reviews, and we appreciate the time you spent on reviewing our paper. In **General Response**, we describe the motivation behind our method and discuss the computational and memory costs (see also the attached **PDF**).
>
> ---
>
> **Q1. Density Estimation.**
>
> Thanks for the note on the density estimation task. Evaluating the density of learned EBM requires the computation of intractable normalization constant (or partition function), which can be computationally expensive and non-exact as observed in [1]. Therefore, we consider standard metrics, such as FID and IS, for the sample quality of our EBM. We also consider out-of-distribution detection for EBM evaluation (please see *Out-of-Distribution Detection*, Section 1.2, Table 3 in Appendix).
>
> **Q2. Biased-learned Generator.**
>
> Our generator model is learned to match the EBM and the true data distribution. In the absence of observed training data, the generator can only learn from the EBM samples, and any approximation error of the EBM may carry over to the generator learning. Consequently, this leads to a biased-learned generator, which, in turn, affects the EBM learning by providing sub-optimal initial points. This is particularly true in the early stage of learning. To demonstrate, we adapt our model and learn the generator without using true data. As shown in Figure 1 in the attached **PDF** such a biased-learned generator might generate noisy and less realistic samples, resulting in sub-optimal EBM training and, consequently, an overall weaker generation performance (see, e.g., cooperative-style baselines in *Image Modelling*, Section 5.1, Table 1).
>
> **Q3. Justification of Langevin Steps.**
>
> The use of Langevin step is intended to refine the direct samples obtained from generator and inference model, guiding them towards target distributions. For example, in L129, the starting (reference) distribution is unimodal Gaussian $q_\phi(z|x)$, using K steps of Langevin would drive such distribution towards the exact generator posterior (i.e., $p_\theta(z|x)$, which is the target distribution of latent MCMC, and can be multi-modal). This is based on the monotonicity property of KL [2], i.e., $KL(T_{\theta_t}^{z}q_{\phi_t}(z|x)||p_{\theta_t}(z|x))\le KL(q_{\phi_t}(z|x)||p_{\theta_t}(z|x))$.
>
> The Figure 2 in Section 5.2 is shown to demonstrate the learned generator and inference model. If they are well-learned and successfully absorb the corresponding MCMC revisions, then they should generate ancestral samples that are on (or lie close to) the major modes of target distributions (or stationary distributions of MCMC transitions). For example, for latent MCMC revision (Figure 2, right panel), the samples exhibit only mild pixel changes along the trajectory, indicating the inference model $q_\phi(z|x)$ is well-learned and close to the stationary distribution, i.e., $KL(T_{\hat{\theta}}^{z}q_{\hat{\phi}}(z|x)||q_{\hat{\phi}}(z|x))\rightarrow 0$ ($\hat{\theta}$ and $\hat{\phi}$ denoted for  learned generator and inference models). This phenomenon (albeit within the realm of image space) is also demonstrated in Section 6.1, Figure 7.b in [3].
>
>
>
> We humbly request that you could reconsider the decision given our response. Please feel free to let us know if you have more questions about the paper. We will try our best to address your concerns. Thank you!
>
> ---
>
> [1] Du, Yilun, and Igor Mordatch. Implicit generation and generalization in energy-based models. Advances in Neural Information Processing Systems, 32, 2019.
>
> [2] Thomas M. Cover and Joy A. Thomas. Elements of Information Theory. John Wiley \& Sons, 1991.
>
>
>
> [3] Jianwen Xie, Zilong Zheng, and Ping Li. Learning energy-based model with variational auto-encoder as amortized sampler. In Proceedings of the AAAI Conference on Artificial Intelligence, 2021

---

> > ### Comment · Reviewer_iX9Y · 2023-08-15
> >
> > I first want to thank the authors for their insightful rebuttal. Based on the rebuttal as well as the feedback provided by the other reviewers, I still have some questions regarding the paper.
> >
> > **The suggestions seem ad-hoc to the reviewers**
> >
> > I agree with reviewers uj7J and Rjbi that many contributions of this paper seem ad-hoc and lack motivation. I believe the main reason is that (as pointed by reviewers uj7J and XcPg) the paper follows the probalistic framework provided by the Divergence Triangle paper. I think that introducing those new ideas by highlighting the contrast with DT (like in the general answer which perfectly highlights the importance of the revised densities) would make the paper much easier to read.
> >
> > **The density estimation task**
> >
> > I like that the authors provided long-run samples from the EBM which suggest a good density estimation. However, I disagree with the fact that the unormalized density is a limitation for density related benchmarks. Not only the unormalized likelihood is sufficient for many benchmarks but looking at a 2D estimate on the checkboard, the spiral distribution or any multi-modal distribution is now a standard in the EBM community.
> >
> > **Biased generator**
> >
> > In your answer, you say that "any approximation error of the EBM may carry over to the generator learning" however isn't the goal of this first MCMC chain to sample the EBM? In other words, it seems important that an approximation error on the EBM should propagate to the generator.
> > I am happy to see that the authors illustrated their point through Fig. 1 of the rebuttal PDF. However, I think that the low resolution of the image as well as the lack of objective metrics don't allow for an accurate comparison.
> >
> > **Langevin sampling**
> >
> > I think there is a miss-understanding of my review here. What I mean is that if the support of $q_{\phi}(z | x)$ is included in the support of one of the modes of $p_{\theta}(z | x)$ then a single Langevin chain initialized with this generator definitely cannot properly mix between the modes of $p_{\theta}(z | x)$ leading to a biased learning. This is exactly what you illustrate in the second part of your answer and in Fig. 2 of the main paper.
> > This lack of proper mixing require many parallel MCMC chains to cover all the modes of $p_{\theta}(z | x)$. This could be fixed by using mixing MCMC such as global-local MCMC algorithms [1,2] (which could leverage the tractable generator as global proposal) as it was recently done in [3] for direct MLE on EBMs.
> >
> > **Conclusion**
> >
> > Overall, I think the motivation of this paper is not very clear as the connection with DT should be highlighted. However, my review indeed seems severe as the authors clearly proved the superiority of their method in a fair and illustrated way. I will increase my score to 5 and I'm ready to increase it again depending on the answer given by the other reviewers or if some density experiments are appended. I think that the discussion on Langevin sampling is a bit out of the scope of this paper (especially if the latent space is not very multimodal which I would like to see).
> >
> > [1] Gabrié, M., Rotskoff, G. M., & Vanden-Eijnden, E. (2022). Adaptive Monte Carlo augmented with normalizing flows. Proceedings of the National Academy of Sciences, 119(10), e2109420119. doi:10.1073/pnas.2109420119
> >
> > [2] Samsonov, S., Lagutin, E., Gabrié, M., Durmus, A., Naumov, A., & Moulines, E. (2022). Local-Global MCMC kernels: the best of both worlds. In S. Koyejo, S. Mohamed, A. Agarwal, D. Belgrave, K. Cho, & A. Oh (Eds.), Advances in Neural Information Processing Systems (Vol. 35, pp. 5178–5193).
> >
> > [3] Grenioux, L., Moulines, É., & Gabrié, M. (2023). Balanced Training of Energy-Based Models with Adaptive Flow Sampling. arXiv preprint arXiv:2306.00684.

---

> > > ### Author Response · Authors · 2023-08-17
> > > **New Response for Questions**
> > >
> > > Thanks for your comments. Please see our answers to the new questions below.
> > >
> > > ---
> > > **Q1. Clarification of Generator Learning**
> > >
> > > Sorry for the confusion. For clarification, we shall begin with a simple analogy of "teacher" (i.e., EBM) and "student" (i.e., generator model) (similar to Section. 1.1 in [c]). In Cooperative Learning [c], the 'teacher' directly learns from the "textbook" (i.e., true data), while the "student" can only learn from the "teacher"s advice.
> > >
> > > In contrast, we learn both the generator and EBM to match the true data distribution; intuitively, the generator model that also fits with observed data should provide a more informative starting point for EBM learning. In the view of analogy, the "student" now has access to the "textbook" and thus can be *"self-learned"* but also needs to take revisions from the "teacher." Our additional experiment in the attached **PDF** (Figure. 1) demonstrates that our generator model can generate better image synthesis than the generator model learned without training observation (FID score at 5K iterations, 39.25 v.s. 58.74).
> > >
> > > **Q2. Langevin Sampling and Density Estimation Task.**
> > >
> > > [We have kindly asked the ACs to forward the **anonymous link** as required by the regulation of NeurIPS. ]
> > >
> > > Thanks for the new references. We agree that MCMC methods, such as HMC or Langevin dynamics, can be challenging in practice to serve as a "convergent" sampler. This is a long-standing problem, as the movement/proposal in each step can be "local" (as indicated by \[1,2]), which in turn results in long-mixing time and ineffective traversal of modes, especially in the realm of high-dimensional multi-modal data. In this work, we mainly focus on the short-run, non-convergent, non-persistent Langevin sampler due to its simplicity, efficiency, and ability to facilitate fair model comparison.
> > >
> > > Such short-run Langevin samplers can be helpful in practice for the learning of realistic samples (see, e.g., *Image Modelling*, Section 5.1) but can also lead to limited EBM density estimation (see 3rd column of *Kernel Density Estimation (KDE) experiment* in Figure. 1 from the **anonymous link**). Such a phenomenon is shared by non-convergent samplers and is studied in [a,b] (see, e.g., Figure. 6 in [a] and Section. 4.1 in [b]). As shown in Table. 5 and Table. 6 in *Ablation Studies*, increasing the steps of Langevin transition during training could render a better sampler, resulting in improved generation/reconstruction quality and density estimation (see also 2nd and 4th column in Figure. 1 from the **anonymous link**).
> > >
> > > The suggested global-local MCMC algorithm is interesting and inspirational. Based on our understanding (sorry in advance if we miss the key points in those papers), our generator can be viewed as an unadjusted "global" proposal to facilitate the "local" Langevin transition (similarly, the inference model shall be viewed as an unadjusted “global” proposal to facilitate the “local” latent posterior Langevin transition). However, different than the NF-based proposal that features tractable log-likelihood, the generator proposal in our work has a lower-dimensional latent space and can be intractable, as the generator is not bijective as those of NF-based models. This can make direct use of the i-SIR framework [2] challenging (e.g., computation of importance weights), since the computation of the marginal generator distribution $p_\theta(x)$ can be typically intractable and costly with the integration of latent variables. The reference [3], along with the related mixing MCMC variants mentioned therein, certainly sheds light on and has the potential to further improve the existing Langevin samplers used in our framework. For example, the local Langevin samplers could be interleaved between global generator proposals to achieve better multi-modal traversal and better negative samples from EBM. We shall leave the deeper exploration along this direction as our future study.
> > >
> > > [Note: we also want to kindly point out that the reference [3], the v1 version, is uploaded to ArXiv **after** our NeurIPS submission].
> > >
> > > Overall, thanks again for such helpful suggestions. We shall cite them in the revision and acknowledge them as effective and generic ways to improve standard MCMC samplers, especially in those complex, multi-modal scenarios.
> > >
> > > ---
> > >
> > > [a] Nijkamp, Erik, Mitch Hill, Tian Han, Song-Chun Zhu, and Ying Nian Wu. "On the anatomy of mcmc-based maximum likelihood learning of energy-based models." In Proceedings of the AAAI Conference on Artificial Intelligence, 2020.
> > >
> > > [b] Nijkamp, Erik, Mitch Hill, Song-Chun Zhu, and Ying Nian Wu. "Learning non-convergent non-persistent short-run MCMC toward energy-based model." Advances in Neural Information Processing Systems 32, 2019.
> > >
> > > [c] Xie, Jianwen, Yang Lu, Ruiqi Gao, and Ying Nian Wu. "Cooperative learning of energy-based model and latent variable model via mcmc teaching." In Proceedings of the AAAI Conference on Artificial Intelligence, 2018.

---

### Official Review · Reviewer_RJbi · 2023-07-06

**Soundness:** 3 good
**Presentation:** 3 good
**Contribution:** 2 fair
**Rating:** 6
**Confidence:** 2

**Summary:**

This work presents a new approach for training energy-based models (EBMs). These are commonly trained by approximating the maximum-likelihood estimate of the model parameters. This is done by minimising a certain Kullback--Leibler divergence via stochastic-gradient descent. To estimate the necessarily gradient, one must approximate an expectation w.r.t. the law of the data under the energy-based model. This is typically done via an MCMC algorithm.

The present work attempts to speed up this procedure by training an additional surrogate model (the "generator" model) from which the MCMC chain can be initialised (so that fewer MCMC iterations are needed). This generator model and the EBM are trained as part of a larger scheme which additionally trains an inference model (i.e. the posterior distribution of the latent variables given the observed data).

-----------------------
EDIT (2023-09-04): I have now read all the reviews and rebuttals. All my questions have been addressed. I continue to be happy for this paper to be published.

**Strengths:**

1. The method appears to be sound and novel.
2. The paper is mostly well written.

**Weaknesses:**

Main comments:

My expertise with training EBMs in this fashion is too limited to offer deeper insights. However:

1. Some of the objectives used for training the three models appear a little ad-hoc and their motivation/justification sometimes leans on convergence of the MCMC chains (which seems unlikely to occur in practice given the small number of MCMC iterations used). At the very least, some added justification/derivation of these objectives would help the reader gain more intuition for the method.
2. I'd ask the authors to confirm that the numerical comparisons are fair in the sense that all algorithms have roughly the same computational cost (see question below).
3. I think it would help to have some (brief/high-level) pseudo code in the main paper.


Minor comments/typos:

L114: grammar in "such Langevin process"
L180: are the KL divergences equivalent to the marginal version or is the /minimisation/ of these KL divergences equivalent to minimising the marginal divergences?
L152--160: I understand what the authors mean with the "transition-kernel" notation. But I think the notation/explanation in this paragraph needs to be improved. For example, in Line 156, the stated expression is referred to as "marginal distribution on $x$" but it clearly represents distribution on joint the space of $(x, z)$. I would state the formal definitions (i.e. those found in Lines 158 and 160) right from the start to enhance clarity. It might also be worth reminding the reader that $p_\theta(x)$ is the marginal of $p(z)p_\theta(x|z)$.
L256--257: the MCMC sampling -> MCMC sampling

- there are sometimes missing spaces between "Eq" and the equation number and between "Sec." and the section number.
- punctuation is needed even if a sentence ends in a (displayed) equation
- the font size in most of the figures is too small (especially in Figure 3) and axis labels are sometimes missing
- it may just be my lack of familiarity with the literature in this area but is it clear to the reader what "explaining away inference" means in this specific context? If not, maybe it'd be worth adding a brief explanation.
- table captions normally go /above/ tables (unless the NeurIPS style guide says otherwise)

**Questions:**

In Table 1 (and also in the numerical results from Sections 5.2 and 5.3), is the computational cost of all algorithms comparable?

---

> ### Author Rebuttal · Authors · 2023-08-08
>
> Thanks for your supportive comments and all the corrections, such as typos, formats, and captions, and we shall correct them in our final version. Please also find a brief description and model comparison in the **General Response** and the attached **PDF**.
>
> ----
>
> **Q1. Convergence of the MCMC Chain.**
>
> The convergence analysis of MCMC chains is mainly used for asymptotically theoretical understanding and is meant to establish the connection with the standard maximum likelihood estimation (MLE) and other relevant models. In practice, it can be costly and even infeasible to run a mixed and convergent MCMC sampler in each iteration. As such, a short-run sampling strategy (i.e., using only a small number of sampling steps) is commonly used and can provide meaningful learning signals for the model traversing as observed in [1]. We add the experiment for the long-run Langevin traversing (with $3,000$ steps, please see Figure 2 in the attached **PDF**), where only minor changes can be observed. It suggests that the generator, though trained with short-run MCMC revision, could essentially learn to match the long-run (i.e., near convergent) MCMC samples.
>
> **Q2. Computation Cost of All Algorithms.**
>
> We follow the standard evaluation protocol in generative modeling (e.g., [2, 3] and others) and report the best-performing results for the baseline models. MCMC-based methods typically involve an inner loop for sampling, which introduces computational overhead compared to variational-based or adversarial methods (see the comparison of computational cost in **General Response** and the attached **PDF**). However, we note that our proposed joint learning scheme allows complementary models (e.g., inference model) to act as informative starting points and jump-start MCMC procedures (e.g., MCMC sampling on generator posterior), making them more efficient and effective than other noise-initialized MCMC methods. For example, as demonstrated in *Analysis of Inference Model* (Section 5.3), the inference accuracy of our 10-step MCMC revision with inference model initializer can be even better than 30-step, noise-initialized MCMC sampling.
>
> **Q3. Pseudo Code, KL Divergences Equivalence, Notation and Expression.**
>
> Thanks for your valuable suggestions. The detailed derivation for the KL divergence equivalence can be found in *Methodology* (Section 2.2 in Appendix). We shall follow your advice to change the notations and expressions (e.g., marginal distribution in L156) in the revision, and we shall open-source our implementation.
>
> **Q4. Explaining-away Inference.**
>
> Sorry for the confusion. The "explaining-away inference" refers to the situation where the knowledge of one latent variable reduces the influence or importance of another latent variable in explaining observed data. We employ this term here to emphasize that during the Langevin process, latent factors compete with each other to explain the given training example. We shall make it more clear in the revision.
>
> ---
>
> [1] Erik Nijkamp, Mitch Hill, Song-Chun Zhu, and Ying Nian Wu. Learning non-convergent non-persistent short-run mcmc toward energy-based model. Advances in Neural Information Processing Systems, 32, 2019.
>
> [2] Zhisheng Xiao, Karsten Kreis, Jan Kautz, and Arash Vahdat. Vaebm: A symbiosis between variational autoencoders and energy-based models. In International Conference on Learning Representations, 2020.
>
> [3] Gao, Ruiqi, Yang Song, Ben Poole, Ying Nian Wu, and Diederik P. Kingma. Learning Energy-Based Models by Diffusion Recovery Likelihood. In International Conference on Learning Representations, 2021.

---

> > ### Comment · Reviewer_RJbi · 2023-08-17
> > **Thank you**
> >
> > I want to thank the authors for their detailed response. I think I'd be happy to see this published.
> >
> > That said, this is somewhat outside my field of expertise (as my low confidence score indicates) e.g., I was unaware of the divergence-triangle paper.

---

> > > ### Author Response · Authors · 2023-08-17
> > >
> > > Thanks for your comments and the time you spent reviewing our paper.
> > >
> > > We are willing to provide further responses if you have more questions about our paper.

---

### Official Review · Reviewer_uj7J · 2023-07-07

**Soundness:** 3 good
**Presentation:** 3 good
**Contribution:** 3 good
**Rating:** 6
**Confidence:** 5

**Summary:**

This work investigates learning EBMs for image data using auxiliary generator and inference models. The model generates samples by drawing a latent normal vector, passing it through the generator, and refining the generator sample using MCMC with the EBM. An inference network, which predicts latent vectors from images, is used to assist with training the EBM and generator. The key innovation is to update both generator samples using image space MCMC and inferred latent vectors for data using latent space MCMC. This allows the generator to be efficiently trained using both revised MCMC samples, as in Cooperative Learning, and revised latent codes for data, which is unique to this work. Training the generator to match both EBM and data samples is crucial for improving performance beyond the results achieved by related methods. Training algorithms for the EBM, generator, and inference network are presented using several KL terms between different joint distributions of images and latent codes. Experiments show strong EBM synthesis results for CIFAR-10, CelebA, CelebA HQ, and LSUN-Church-64.

**Strengths:**

* The work identifies and addresses a key limitation of the Cooperative Learning framework, which is the fact that the generator is trained to reconstruct only refined EBM samples. This can limit the potential of the generator since EBM samples are less realistic than true data. While other works such as Divergence Triangle train the generator to reconstruct data images using inferred codes for observed data, these works lack an MCMC refinement stage which is crucial for improving sample quality. This work bridges the gap between Cooperative Learning style methods and Divergence Triangle style methods to incorporate both MCMC sampling and generator data reconstruction into a single learning framework.
* Experimental results show significant benefits from the proposed method compared to a variety of other EBM learning methods.

**Weaknesses:**

* The loss functions appear to be significantly inspired by the Divergence Triangle, but the paper lacks a thorough comparison between the proposed method and Divergence Triangle. Such a discussion would help elucidate the differences between the proposed method and prior work. To my understanding, the primary differences are 1) replace all LHS of KL terms with distributions defined by short-run MCMC distributions with frozen model parameters, 2) add KL($\tilde{P} || P$) and KL($\tilde{Q} || Q$) terms to the generator and inference network updates respectively.
* The motivation of each loss term is not entirely clear, which makes the proposed loss gradients appear somewhat ad-hoc. An intuitive explanation of the role of each KL term would be very helpful for the reader. A clearer discussion of connections to the Divergence Triangle could also help with understanding the KL terms.
* The proposed method is somewhat complex, and the loss functions are defined sequentially given the current fixed model weights. Such an approach is common to Cooperative Learning and related methods, while amortized methods like Divergence Triangle have a loss function which can be written in closed form without reference to fixed model weights. While this is not a major issue in my view, the reliance on loss defined by a sequence of fixed model weights as opposed to a single loss function should be discussed as a limitation of the proposed method.

Minor Problems
* The related work [12] uses MCMC sampling initialized from the generator to update the EBM similar to [36, 37], instead direct generator samples like [11, 9, 19].

**Questions:**

Could the authors provide a detailed comparison between their proposed method and the Divergence Triangle?

**Limitations:**

A broader impact section was not included. Computational limitations are very briefly discussed, but more thorough details about runtime and compute costs compared to related methods would strengthen the paper.

---

> ### Author Rebuttal · Authors · 2023-08-08
>
> Thanks for your insightful feedback. We appreciate the correction of the references and shall fix them in our final manuscript.
>
> We provide the clarification in **General Response** for
> - Connection with Divergence Triangle (**G.2**)
> - Motivation of our method (**G.1**)
> - Computational cost (**G.3**) (see also the attached **PDF**)
>
> ----
>
> **Q1. Closed-form Objective.**
>
> You are correct. Our MCMC teaching-based framework, indeed, cannot be learned within a single objective function. The MCMC revision shall be conducted based on the fixed model weights. We shall include this point and a thorough discussion of the limitation in our revision.

---

> > ### Comment · Reviewer_uj7J · 2023-08-17
> > **Thanks for the responses**
> >
> > Thanks to the authors for their responses. I have read the responses to myself and other reviewers and find that they provide satisfactory answers. I will keep my score.

---

> > > ### Author Response · Authors · 2023-08-17
> > >
> > > We thank you for your valuable comments.
> > >
> > > Please let us know if you have any further questions, and we are willing to make further clarification.

---

### Author Rebuttal · Authors · 2023-08-08

We thank the valuable comments from all the reviewers. The reviewers note that (1) our idea is *novel* (**Reviewer** **uj7J**, **RJbi**, **XcPg**), (2) our experiments are *clear* and *solid* (**Reviewer** **uj7J**, **iX9Y**, **XcPg**), (3) our paper is *well-written* (**Reviewer RJbi**, **XcPg**). We are also encouraged to hear that you found our paper addressed the *key limitation* and *actual problem* for future research (**Reviewer uj7J,** **XcPg**).

We provide the general response for the common questions below and respond to each reviewer in their individual comments.

We refer to the attached **PDF** for additional ablation results.

----

### G.1 Motivation (Reviewer **uj7J, RJbi, iX9Y, XcPg**).

**Dual-MCMC Teaching.** This paper studies the learning of the energy-based model (EBM). The EBM can be typically learned through the MLE, which involves MCMC sampling and is known to be challenging in practice. To tackle this challenge, we introduce a probabilistic learning framework, where a generator and inference model are jointly trained with the EBM.

Firstly, our joint learning scheme is designed to facilitate the intractable EBM and generator learning, and both generator and inference models are learned to jump-start MCMC samplings for EBM and generator posterior distribution, respectively. Such a joint learning scheme is related to work **Divergence Triangle** [1] (as noted by **Reviewer** **uj7J**, **XcPg**). However, DT relies on ancestral samples from both the generator and inference model, which may be sub-optimal. Additionally, the learning can be fundamentally different (please refer to **G.2** for a detailed breakdown and comparison). Secondly, the KL terms used in generator and inference model learning are designed to integrate two MCMC revised samples (one in the latent space and another in the image space) to steer the generator and inference learning (as highlighted by **Reviewer uj7J**). Consequently, this approach would enhance EBM learning by providing improved initial samples for MCMC samplings. We shall add a detailed discussion of the motivation in our final version.

### G.2 Connection to Divergence Triangle (DT) [1] (Reviewer **uj7J, XcPg**).

We denote $Q$ for $Q_{\phi}(x, z)$ and other joint densities for notation simplicity. Recall that $Q=q_\phi(z|x)p_{\rm data}(x)$,$P=p_\theta(x|z)p(z)$, $\Pi=\pi_\alpha(x)q_\phi(z|x)$, and $\tilde{Q}=p_{\rm data}(x)T^z_{\theta_t} q_{\phi_t}(z|x)$, $\tilde{P}=T^x_{\alpha_t} p_{\theta_t}(x|z)p(z)$, where $T^z_{\theta_t}(\cdot)$ denotes the Markov transition kernel of finite step Langevin dynamics that samples $z$ from $p_{\theta_t}(z|x)$, and $T^x_{\alpha_t}(\cdot)$ denotes the transition kernel that samples $x$ from $\pi_{\alpha_t}(x)$.

- For learning the EBM ($\alpha$), we use $\min_\alpha KL(\tilde{Q}\| \Pi) - KL(\tilde{P}\|\Pi)$  (Eqn. 7), and DT considers $\min_\alpha KL(Q\| \Pi) - KL(P\|\Pi)$. Specifically, we utilize the MCMC revised samples (i.e., joint densities of $\tilde{Q}$ and $\tilde{P}$) for learning the EBM, while DT only considers ancestral samples (i.e., joint densities of $Q$ and $P$). Such MCMC-revised samples should render more effectively learned EBM, as model samples revised by the EBM itself can be more accurate than samples directly drawn from the generator, i.e., $KL(T^x_{\alpha_t}p_{\theta_t}(x)\|\pi_{\alpha_t}(x))\le KL(p_{\theta_t}(x)\|\pi_{\alpha_t}(x))$.
- For learning the generator model ($\theta$), the KL terms for our work are $\min_\theta KL(\tilde{Q}\| P) + KL(\tilde{P}\|P)$ (Eqn. 10), and DT's are $\min_\theta KL(Q\|P) + KL(P\| \Pi)$. In comparison, for training observation, our generator is learned with revised latent samples ($KL(\tilde{Q}\| P)$ v.s. $KL(Q\|P)$). Such samples can be more accurate towards the generator posterior, i.e., $KL(T^z_{\theta_t}q_{\phi_t}(z|x)\|p_{\theta_t}(z|x))\le KL(q_{\phi_t}(z|x)\|p_{\theta_t}(z|x))$. For generated model samples, our model, i.e., $KL(\tilde{P}\|P)$, learns to match the revised MCMC samples from $\pi_\alpha(x)$, while DT intends to chase the major modes of $\pi_\alpha(x)$ through variational approximation (i.e., $KL(P\| \Pi)$).
- For learning the inference model ($\phi$), we utilize $\min_\phi KL(\tilde{Q}\| Q) + KL(\tilde{P}\|\Pi)$ (Eqn. 15), and DT uses $\min_\phi KL(Q\|P) + KL(P\|\Pi)$. Two learning schemes can be very different. On the one hand, for training observation, our model, i.e., $KL(\tilde{Q}\| Q)$, is trained by amortizing the latent MCMC, while the DT follows variation inference as in VAEs (i.e., $KL(Q\|P)$). On the other hand, on generated model samples, our inference is learned to match the revised generator samples from $\pi_\alpha(x)$ (i.e., $KL(\tilde{P}\|\Pi)$), while the DT directly considers ancestral generator samples, i.e., $KL(P\|\Pi)$, which can be sub-optimal.

### G.3 Computational and Memory Cost (Reviewer **uj7J, RJbi, iX9Y, XcPg**).

Our learning algorithm belongs to MCMC-based methods and can indeed incur computational overhead due to its iterative nature compared to variational-based or adversarial methods. We provide further analysis by computing the wall-clock training time and parameter complexity for our related work Divergence Triangle [1] (variational and adversarial-based joint training without MCMC) and our model (see Table 1 in the attached **PDF**), where the proposed work requires more training time but can also render significantly better performance. In terms of memory cost, it's important to note that we didn't observe further improvement by just increasing parameter complexity (see *Parameter Efficiency* in Section 1.1 of the Appendix). This emphasizes the effectiveness provided by our learning algorithm.

---

[1] Tian Han, Erik Nijkamp, Xiaolin Fang, Mitch Hill, Song-Chun Zhu, and Ying Nian Wu. Divergence triangle for joint training of generator model, energy-based model, and inferential model. In Proceedings of the IEEE Conference on Computer Vision and Pattern Recognition, 2019

---

### Decision · Program_Chairs · 2023-09-21

**Decision:**

Accept (poster)

**Comment:**

The paper presents a significant advance in the field of energy-based models (EBMs). The authors identify and tackle existing limitations in Cooperative Learning frameworks, particularly focusing on the generator's constraint in only being able to reconstruct refined EBM samples.

The proposed method effectively bridges existing methods, integrating elements from both Cooperative Learning and Divergence Triangle techniques to create a comprehensive framework that features MCMC sampling and generator data reconstruction.

The results from the experiments carried out provide strong support for the proposed methodology, indicating its advantages over existing EBM learning methods. Furthermore, the authors were engaging and responsive during the rebuttal period, providing reproducible code and clear theoretical explanations, which were well received by the reviewers.